# Gender and Anthropometric Effects on Seat-to-Head Transmissibility Responses to Vertical Whole-Body Vibration of Humans Seated on an Elastic Seat

**Yumeng Yao [1],\*, Krishna N. Dewangan [2] and Subhash Rakheja [3]**

[1] School of Mechanical Engineering, University of Shanghai for Science and Technology, Shanghai 200093, China

[2] Department of Agricultural Engineering, North Eastern Regional Institute of Science and Technology, Nirjuli 791109, India

[3] CONCAVE Research Center, Concordia University, Montreal, QC H3G 1M8, Canada

\* Correspondence: yu_yao@usst.edu.cn

**Abstract:** This study investigated the effects of gender and ten different anthropometric parameters on the vertical vibration transmission from seat to the head of the body seated on an elastic seat. The seat-to-head transmissibility (STHT) responses in the vertical and fore-aft directions of 58 participants (31 males and 27 females) were measured under three levels of vertical vibration (root mean square acceleration: 0.25, 0.50, and 0.75 m/s$^2$) in the 0.50–20 Hz range, when sitting on a viscoelastic seat with and without a vertical back support, and with hands on a steering wheel. Apart from the important effects of elastic coupling between the body and seat, the results show distinctly different vertical and fore-aft STHT responses from the two genders. Moreover, the gender effect was strongly coupled with back support and excitation conditions. The primary resonance frequencies of male subjects were higher than those of female subjects, while the peak vertical STHT magnitudes were comparable. Owing to the strong coupled effects of gender and anthropometric dimensions, the study is designed to reduce the coupling by considering datasets for subjects with comparable chosen dimensions. Among the various anthropometric dimensions considered, the body mass and fat mass revealed strong influences on the primary resonance frequency, which was similar for male and female subjects with comparable body mass index and body fat mass. The vertical STHT magnitude of the two genders with the same lean body mass was also nearly identical. The peak fore-aft STHT magnitudes of the male subjects were notably higher than those of the female subjects with comparable anthropometric dimensions with the exception of the body mass.

**Keywords:** whole-body vibration; seated body biodynamic response; STHT response; vibration transmissibility; elastic seat; gender effect; body mass effect; body fat effect





## 1. Introduction

The transmission of vertical seat vibration to the seated human occupants' head have been experimentally investigated under different ranges of experimental conditions to gain a better understanding of human responses to whole-body vibration (WBV). The measure, often expressed by the seat-to-head vibration transmissibility (STHT), is generally obtained for the human occupant seated on a rigid seat, while considering broad ranges of magnitude and frequency of seat vibration, sitting postures, and hands position [1–4]. The underlying reason for the usage of a rigid seat was to attain biodynamic responses to vibration uncoupled from the seat. This condition, however, does not represent a vehicle seat, which invariably exhibits viscoelastic properties. It has been shown that sitting on a flexible seat yields more uniform body-seat contact pressure compared to that observed on a rigid seat [5,6]. Mean and peak ischium pressures obtained with elastic seats are substantially lower than those with rigid seats [7]. The body coupling with a flexible

seat can thus significantly alter the biodynamic responses of the body to seat vibration. Hinz et al. [8] and Dewangan et al. [9] measured apparent mass (APMS)characteristics of human subjects seated on rigid as well as flexible seats under vertical WBV. Both the studies showed important influences of flexible seats on vertical APMS responses of the seated human.

Flexibility of the seat also affects the transmission of seat vibration to the head, although only a few studies have reported STHT responses with an elastic seat. The vertical seat vibration transmitted to L3 and head of human subjects seated on elastic seats was reported in [10]. The study employed impact type of vertical excitation, and three female subjects seated on three different seats, although the stiffness properties of the seats were not presented. The study showed notable differences in peak STHT magnitudes and resonance frequencies observed with the elastic and rigid seats. Dewangan et al. [11] also reported considerable differences in vertical and fore-aft STHT responses measured with rigid and elastic seats subject to vertical WBV. The study performed measurements with a total 58 human subjects coupled with a rigid seat and three different elastic seats, exposed to broad band vertical vibration in the 0.5 to 20 Hz frequency range. The peak magnitudes and primary resonance frequencies with the elastic seats were significantly different from those obtained for the rigid seat.

Apart from the elastic properties of the seat, the STHT responses to vibration are strongly influenced by various anthropometric dimensions. A comparison of STHT responses reported in different studies suggested considerable differences among them [12]. Similar differences were also observed in a synthesis of selected STHT datasets [3]. The observed differences among the reported data together with large inter-subject variabilities in individual studies are partly caused by variations in physical characteristics of the subjects, such as body mass and anthropometric dimensions. The male and female populations, in particular, exhibit broad differences in their physical characteristics. The majority of the anthropometric dimensions of an average male are larger than those of an average female. Morphologically, the skeleton of a male is generally considerably larger than that of a female, while the pelvic region of a female is relatively delicate and round [13]. The pelvis of a female is relatively short and wide, while that of a male is tall and narrow. The females thus exhibit relatively greater hip width and thereby higher body–seat contact area and lower mean contact pressure [9,14]. The trunk size and the lumbar curvature as well as tilt also differ between the two genders [15,16]. Moreover, males possess higher muscle mass than females, while the fat mass is greater for females. Body fat is mostly accumulated in the thighs and hips among females, while it is accumulated around the waist in males. Owing to such differences in physical characteristics, the transmission of seat vibration to the head may differ for the male and female subjects.

Studies reporting STHT characteristics of body seated on rigid seats have mostly focused on male subjects [4,17,18], while the effects of gender have been addressed in relatively fewer studies [14,19]. The effect of gender on the STHT responses, however, could not be clearly established due to large anthropometric differences between the two gender groups [20]. The STHT data obtained for the two gender groups suggest relatively higher primary resonance frequencies for males compared to females [14]. Goggins et al. [19] assessed seat-to-head transmissibility and self-reported discomfort of five females and six males considering different sitting postures, while the gender effect on STHT responses was not investigated.

Since vibration transmits from the seat to the head, the mass, build, and dimensions of the upper limb of the body may be important influencing parameters. Only a few studies, however, have attempted to explore the effects of anthropometric parameters on the STHT responses. Donati and Bonthoux [21] studied vibration transmissibility from the seat to the thorax and pelvis and its correlations with seated body mass and selected body dimensions such as stature, trunk height, sitting height, and chest circumference. The STHT magnitudes at frequencies up to 4 Hz were higher for subjects with higher, while the STHT magnitudes at higher frequencies were negatively correlated with seated body

mass and some of the thorax dimensions. Dewangan et al. [14] investigated the effects of 10 anthropometric parameters grouped under mass-, build-, and stature-related parameters to study the effect of body dimensions on the STHT responses with rigid seats. The data obtained with both gender groups showed strong effects of body mass, while the gender effect was reported to be coupled with various anthropometric parameters.

In the aforementioned studies, the transmission of vertical seat vibration to the head of seated human subjects have been limited to rigid seats for the purpose of characterizing biodynamic responses of the body alone to vibration. In the rigid seating condition, the body mass is concentrated in a relatively smaller contact area of the seat in the vicinity of the ischial tuberosities. Flexibility of typical seat cushion, however, causes the seated body mass to distribute more uniformly over a larger area of the seat [5–7,9]. The vertical vibration biodynamic responses measured with flexible seats thus differ from those obtained with a rigid seat. Moreover, the gender effects on STHT, measured with rigid seats, could not be clearly established, likely due to vast differences in anthropometric dimensions of the two gender groups. The influences of anthropometric dimensions and gender on the STHT responses measured with flexible seats have not yet been investigated. The gender effect could be more accurately assessed by considering male and female subject samples with somewhat comparable anthropometric parameters.

In the present study, the vertical and fore-aft STHT data, acquired with 58 (31 males and 27 females) subjects seated on an elastic seat with and without the back support, are analyzed in an attempt to investigate the effects of different anthropometric dimensions and the gender. The data considered in the study were acquired under three levels of vertical vibration of the seat (rms acceleration: 0.25, 0.5, and 0.75 m/s$^2$) in the 0.5–20 Hz frequency range. The test subjects were grouped to obtain sub-samples of male and female subjects with comparable anthropometric parameters relevant to body mass, build and stature. The subjects were subsequently grouped so as to achieve comparable body mass and anthropometric dimensions of male and female participants in order to establish gender effect on the vertical and fore-aft STHT responses to vertical WBV.

## 2. Measurement and Analysis Methods

### 2.1. Subjects and Seat Characteristics

Fifty-eight (31 males and 27 females) subjects with a wide range of anthropometric dimensions were recruited for the experiments designed for measuring vertical and fore-aft STHT responses to vertical seat vibration. All the selected subjects were healthy with no signs of any musculoskeletal disorders including back pain. Each subject was informed about the experimental setup, measurement methods, and safety guidelines prior to the experiments, which were approved by the concerned Human Research Ethics Committee of Concordia University, where the experiments were conducted. Various anthropometric parameters of the participants were measured. These were subsequently grouped in mass-related (body mass, body mass index, and body fat and lean masses), build-related (hip circumference, seat pan contact area and mean contact pressure), and stature-related (overall height, seated height, and cervical vertebrae (C7) height) parameters. Table 1 summarizes the means and ranges of selected anthropometric dimensions of male and female subjects together with the standard deviations. The body fat and lean masses, presented in Table 1, were estimated using the US Navy formula [22].

An elastic seat comprising a flat 8 cm thick polyurethane foam (PUF) cushion with leather covering was selected for the study (Figure 1). The static force-deflection property of the seat stiffness was measured, where the force was applied to the seat via a 200 mm diameter loading indenter recommended in SAE J1013 (200711, Society of Automotive Engineers, Pittsburgh, PA, USA) [23]. The procedure for measurements of the static stiffness is given in detail by Dewangan et al. [11]. Measured force-deflection data revealed progressively increasing stiffness characteristics of the elastic seat with the applied load. The static stiffness corresponding to 330, 440, and 530 N applied loads were obtained as 20.6, 34.0, and 40.7 kN/m, respectively. Wu et al. [5] reported that 48–61% of the standing body mass

is supported by the ischium region of the human buttock. Assuming that 55% of standing body mass is supported by the ischium region on average, the aforementioned loads would correspond to standing body masses in the vicinity of 60, 80, and 96 kg, respectively.

**Table 1.** Mean, standard deviation, minimum, and maximum values of the selected anthropometric parameters of the participants.

| Particulars | | Minimum, Maximum, Mean (Standard Deviation) | |
|---|---|---|---|
| | | Male (*n* = 31) | Female (*n* = 27) |
| | Age (years) | 23.0, 58.0, **31.2** (7.2) | 19.0, 49.0, **36.0** (7.1) |
| Mass-related | Body mass (kg) | 55.0, 106.0, **79.8** (15.7) | 45.5, 72.5, **60.1** (8.3) |
| | Body mass index (kg/m$^2$) | 19.96, 34.99, **26.12** (4.24) | 15.78, 26.31, **22.52** (2.73) |
| | Body fat (%) | 16.10, 37.72, **23.59** (5.93) | 19.26, 39.06, **30.53** (4.83) |
| | Body fat (kg) | 10.5, 39.0, **19.8** (8.2) | 8.8, 25.3, **18.6** (4.7) |
| | Lean body mass (kg) | 43.3, 77.5, **61.6** (9.0) | 34.1, 49.5, **41.6** (4.8) |
| Build-related | Hip circumference (cm) | 88.0,116.0, **103.6** (7.4) | 89.5, 109.0, **99.9** (5.5) |
| Stature-related | Stature (m) | 1.59, 1.92, **1.75** (0.08) | 1.48, 1.73, **1.63** (0.07) |
| | Sitting height (cm) | 81.3, 96.7, **88.8** (6.2) | 63.2, 88.3, **81.0** (7.7) |
| | C-7 height (cm) | 59.4, 74.4, **66.2** (4.6) | 56.5, 70.4, **61.4** (4.2) |

*n*: number of subjects; bold: all means.

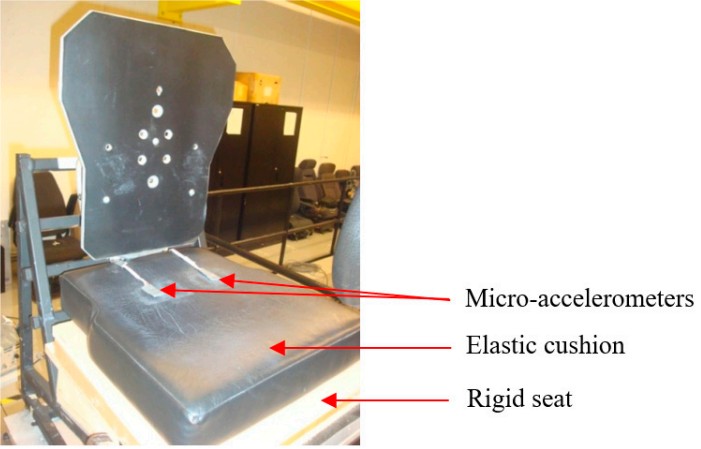

**Figure 1.** Elastic cushion used in the experiment and mounting of accelerometers on the cushion.

### 2.2. Experimental Setup and Measurement Methods

The measurements of vertical STHT responses were performed on a whole-body vertical vibration simulator (WBVVS) described in [24]. The WBVVS was comprised of a platform with a rigid seat pan, a steering wheel and a steering column. The platform was mounted on two vertical servo-hydraulic actuators, which permitted synthesis of desired vertical vibration via a closed-loop vibration controller. The seat cushion was installed on the rigid seat pan, while the backrest was kept rigid. The height of the seat surface from the platform was 445 mm. The hands supported on the steering sub-permitted a driving-like sitting posture. The experimental methods have been described in details in [9]. Briefly, a tri-axial accelerometer mounted in a lightweight helmet-strap mounting system, described by Wang et al. [25], was used to record the vertical and fore-aft motions of the head. A ratchet mechanism was provided in the helmet strap to adjust its tension to fit different subjects. The vertical vibration of the platform was measured via single-axis accelerometer (B&K 4370). The vibration at the driver-seat interface in the vicinity of ischial tuberosities were measured using two micro-accelerometers (ADXL 330) fixed

on the cushion surface (Figure 1). The validity of the body-seat interface accelerometers was established by comparing the responses with those of the standardized seat-pad accelerometer, as reported by Dewangan et al. [9].

It was recognized that the elastic seat significantly alters the platform vibration transmitted to the seated body. Moreover, the vibration transmission property of an elastic seat is nonlinearly dependent on the seated body mass. The acceleration data acquired at the WBVVS platform and seat surface were analyzed to obtain the vibration transmissibility magnitude of the seat for each participant and test condition. The measured transmissibility characteristics showed notable dependence on the seated body mass and excitation magnitude. As an example, Figure 2 illustrates vibration transmissibility characteristics of the elastic seat for both genders with NB support and exposed to 0.50 m/s$^2$ random white noise excitation in the 0.5–20 Hz frequency range. Relatively large inter-subject variabilities are evident for both genders, suggesting strong dependence of STHT on the body mass and anthropometric parameters. The mean transmissibility characteristics of the two gender groups, however, are comparable, as seen in Figure 2c. The experiments were thus designed to ensure identical controlled vibration at the body–seat surface for each subject, which was realized via a vibration controller (Vibration Research Corporation, Jenison, MI, USA). The mean of the two cushion-mounted accelerometers served as feedback to the vibration controller. Three different levels of white noise vibration (0.25, 0.50, and 0.75 m/s$^2$) at the body–seat interface were synthesized in the 0.5–20 Hz frequency range with nearly flat acceleration PSD. A total of 9 drive files (3 excitations × 3 body masses) were synthesized considering three different body masses, namely, 55, 81, and 96 kg. These were considered to represent the range of body masses of the test subjects and ensured nearly identical desired excitations at the body–seat interface for all the test subjects.

The helmet-strap head accelerometer was positioned on the head of the participant sitting upright on the seat cushion without the back support and hands on the steering wheel. The orientation of the head accelerometer was adjusted by the experimenter and the subject was asked tightened the strap using the ratchet mechanism. It was ensured that the helmet-strap was comfortable for the subject as well as sufficiently tight. Additional feet support was used to ensure that the subject's knees were bent about 90° with thighs parallel to the platform and lower legs perpendicular to the platform. The sitting posture and positioning of the head accelerometer was monitored by the experimenter during the trial. In order to minimize the potential orientation error of the head accelerometer during a trial, the subject was instructed to gaze at a target fixed on a wall facing the WBVVS at the eye level. The WBVVS was operated to achieve a desired level of vibration, and the signals from the cushion, seat base and head accelerometers were acquired in a multi-channel spectral analyzer (B&K PULSE 11.0). The bandwidth was fixed as 50 Hz with 800 spectral lines. Each measurement was repeated two times. Each measurement, included two trials, lasted about 4 min including 2 min rest period between the trials.

### 2.3. Data Analysis

The vertical and fore-aft STHT magnitude and phase responses were obtained for each participant and test condition (excitation, sitting posture) using the $H_1$ frequency response function following the Hanning window time averages and 75% overlap of 75%. Owing to the important effects of body mass on the biodynamic responses reported for rigid seats [9,17], the datasets for all the male and female subjects were initially grouped into three different body mass ranges (males: 55–65, 75–85, and 90–106 kg; females: 45–55, 55–65, and 66–72.5 kg). Nine subjects were included in each mass range, as shown in Table 2. The mean masses of the three groups of male subjects were 61.0 ± 4.3, 81.6 ± 4.1, and 96.7 ± 6.4 kg, while for the female subjects, these were 50.4 ± 3.3, 61.0 ± 2.8, and 69.1 ± 2.7 kg (Table 2). In order to study the of selected anthropometric parameters on the STHT response characteristics, the data acquired with male and female subjects were further grouped into three different ranges of each of the selected anthropometric dimension. The selected ranges were established on the basis of measured dimensions of the

subjects, where the number of subjects varied from 6 to 11. The mean, number of subjects, and standard deviation of the selected anthropometric parameters for the three ranges for the male and female subjects are presented in Table 2.

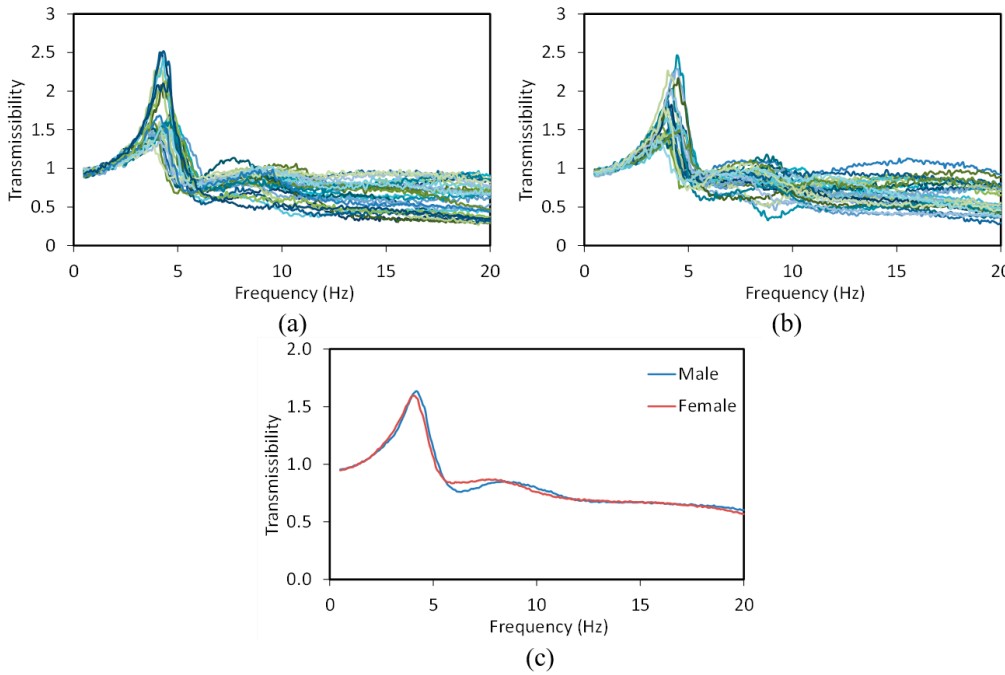

**Figure 2.** Acceleration transmissibility of the seat with 31 males and 27 females under 0.50 m/s² vertical vibration at the body–seat interface: (**a**) male subjects; (**b**) female subjects; and (**c**) mean (without back support).

The grouping of data showed that the mean body mass and anthropometric parameters of each of the three groups for the male and female subjects differ considerably. In order to uncouple the effect of body mass and other anthropometric parameters on the STHT responses, further grouping of the datasets were undertaken to obtain comparable values of body mass and selected anthropometric dimensions of males and females. This, however, involved some complexities due to considerably higher mass of (55 to 106 kg) compared to that of females (45.5 to 72.5 kg). The two gender groups also showed large differences in their anthropometric dimensions, which further contributed complexity of the grouping task. From the measured dimensions of the participants, comparable ranges of anthropometric dimensions could be identified for relatively smaller sample sizes of males and females, which are summarized in Table 2. The task also permitted datasets for 7 males and 7 females with body mass in the vicinity of 60 and 70 kg, which could facilitate a more reliable analysis of the gender effect considering comparable body mass. These subsets for for each of the two mass ranges are denoted as 'G60' (males: 60.4 ± 4.2 kg; females: 61.0 ± 2.6 kg) and 'G70' (males: 70.3 ± 3.7 kg; females: 69.6 ± 2.7 kg). Similarly, 6 to 11 subjects representing comparable values for each anthropometric dimension for the male and female subjects were identified, which are presented in Table 2. The data acquired for the subjects within the identified subgroups were subsequently analyzed for the gender effect on the basis of mean responses of each subgroup with comparable selected anthropometric parameters and the body mass. The statistical significance of main factors, namely, gender, back support, and excitation magnitude, were evaluated using three-way analyses of variance (ANOVA). The data subsets for two gender groups with comparable ranges of selected anthropometric dimensions (Table 2) were further analyzed to evaluate the gender effect via a paired *t*-test. The paired *t*-test was also performed for datasets with the two back support and three excitation magnitudes.

**Table 2.** Means (standard deviations) of selected anthropometric dimensions of males and females subsets corresponding to three comparable ranges of body mass and anthropometric dimensions.

| Gender | Male | | | Female | | |
|---|---|---|---|---|---|---|
| **Anthropometric Parameters** | **Mean (*n*, Standard Deviation)** | **Range** | ***n*** | **Mean (*n*, Standard Deviation)** | **Range** | ***n*** |
| Body mass (kg) | 61.0 (9, 4.3); 81.6 (9, 4.1); 96.7 (9, 6.4) | 55–65 / 75–85 / 90–106 | 9 / 9 / 9 | 50.4 (9, 3.3); 61.0 (9, 2.8); 69.1 (9, 2.7) | 45–55 / 55–65 / 66–72 | 9 / 9 / 9 |
| Body mass (kg)—G60 | - | 55.0–65.0 | 7 | - | 57.0–65.0 | 7 |
| Body mass (kg)—G70 | - | 66.0–75.0 | 7 | - | 66.0–72.5 | 7 |
| BMI (kg/m$^2$) | 21.6 (11, 1.0); 25.4 (11, 1.6); 31.4 (8, 2.0) | 23.3–27.5 | 11 | 19.4 (9, 1.5); 22.6 (8, 0.8); 25.3 (10, 0.7) | 24.4–26.3 | 10 |
| Body fat (kg) | 11.0 (11, 1.6); 16.6 (10, 2.2); 26.2 (7, 3.3) | 19.0–29.0 | 7 | 13.5 (8, 1.1); 19.1 (9, 1.5); 23.7 (9, 1.3) | 21.5–25.3 | 9 |
| Body fat (%) | 16.6 (9, 2.4); 21.9 (10, 1.0); 28.9 (6, 1.6) | 26.9–31.2 | 6 | 25.9 (9, 0.8); 30.8 (9, 2.1); 35.7 (9, 1.7) | 27.9–33.8 | 9 |
| Lean body mass (kg) | 50.1 (9, 4.7); 61.5 (10, 2.4); 68.8 (11, 3.8) | 43.3–54.5 | 6 | 36.0 (8, 1.2); 41.4 (11, 2.1); 47.3 (8, 1.8) | 45.4–49.5 | 8 |
| Hip circumference (cm) | 95.5 (9, 2.0); 102.8 (11, 2.8); 110.7 (9, 4.0) | 98.3–106.4 | 11 | 92.6 (8, 1.9); 100.4 (9, 2.2); 105.2 (10, 1.6) | 97.0–103.0 | 9 |
| Contact area (cm$^2$) | 362 (10, 69); 556 (8, 34); 666 (8, 30) | 615–695 | 8 | 350 (9, 63); 510 (9, 48); 682 (6, 68) | 600–760 | 6 |
| Mean peak pressure (kPa) | 9.1 (11, 0.9); 13.2 (10, 1.0); 17.1 (8, 1.9) | 8.1–10.4 | 11 | 7.2 (9, 1.0); 9.3 (9, 0.5); 12.3 (8, 1.4) | 8.7–10.2 | 9 |
| Stature (m) | 1.70 (9, 0.02); 1.75 (10, 0.02); 1.81 (8, 0.02) | 1.60–1.72 | 10 | 1.56 (9, 0.05); 1.64 (9, 0.02); 1.71 (9, 0.01) | 1.61–1.67 | 9 |
| Sitting height (cm) | 86.1 (8, 1.5); 91.2 (10, 1.6); 95.2 (8, 1.1) | 83.0–87.5 | 8 | 80.4 (8, 1.7); 84.4 (8, 1.0); 88.3 (8, 1.2) | 83.0–85.5 | 8 |
| C7 height (cm) | 61.9 (9, 1.7); 67.4 (10, 1.0); 71.0 (8, 1.5) | 65.8–68.7 | 10 | 58.0 (8, 1.2); 61.4 (8, 0.9); 65.1 (8, 1.7) | 63–67.6 | 8 |

*Note: row labels "Mass-related", "Build-related", "Stature-related" group the rows as in the original left-hand column.*

*n*: number of subjects.

## 3. Results

### 3.1. Gender Effect

As an example, Figure 3a,b show the vertical STHT magnitude responses of 31 males and 27 females, respectively, with NB sitting condition and 0.50 m/s$^2$ RMS acceleration excitation. The results attained for both the gender groups revealed large inter-subject variability. The peak response magnitude for the males occurred in a relatively wider frequency range (4.13–6.38 Hz), while that for the females was observed in smaller frequency range (4.00–5.31 Hz). Moreover, the ranges of peak magnitudes differed notably for the two gender groups. The peak STHT magnitudes for the males ranged from 1.7 to 2.9, while

those for the females occulted in the.97 to 3.15 range. The coefficient of variation (CoV) of the peak magnitude was considerably greater for the male subjects (16–38%) compared with the female subjects (14–20%) for the NB sitting condition. The addition of the back support, however, helped reduce the scatter in the measured responses. For the WB condition, the corresponding CoVs ranged from 15 to 30% for males, and from 10 to 20% for females. A similar degree of variability was also evident in the phase responses. The analyses hereafter, however, are limited to magnitude responses alone. The fore-aft STHT magnitude responses, shown in Figure 3c,d, revealed considerably higher inter-subject variabilities for both gender groups (CoV: 28–38% for males, and 35–41% for females). Unlike the peak vertical STHT magnitudes, the peak fore-aft STHT magnitudes were consistently higher for males (1.89–4.51) compared to females (1.23–4.01). The peak fore-aft magnitudes occurred in somewhat comparable frequency ranges for the male (3.50–6.00 Hz) and female (3.50–5.69 Hz) subjects. Similar variabilities were also observed in the responses obtained under other test conditions.

The mean responses of the male and female subjects were subsequently evaluated for the two sitting and three excitation conditions. As an example, Figure 4a,b illustrate the mean vertical and fore-aft STHT magnitude responses, respectively, of 31 males and 27 females for the NB and WB sitting conditions and 0.50 m/s$^2$ RMS acceleration excitation. The peak magnitude of mean vertical STHT for the female subjects is higher than that obtained for the male subjects, while the primary resonance frequency for female subjects is slightly lower, irrespective of the sitting condition. The mean vertical responses exhibit a distinct secondary peak near 10 Hz in the presence of the back support, which is not evident for the NB condition. The female subjects exhibited a higher magnitude near this secondary resonance as compared with the male subjects. An opposite gender effect, however, is observed in the mean fore-aft STHT responses. The mean fore-aft peak magnitude in the vicinity of the primary resonance frequency response ($\approx$5 Hz) is considerably higher for males compared to females for the NB sitting condition. The fore-aft peak magnitudes for both gender groups, however, are comparable for the WB condition, although males exhibit a higher magnitude near the secondary resonance for both sitting conditions. Similar trends were also observed in mean responses obtained for other test conditions.

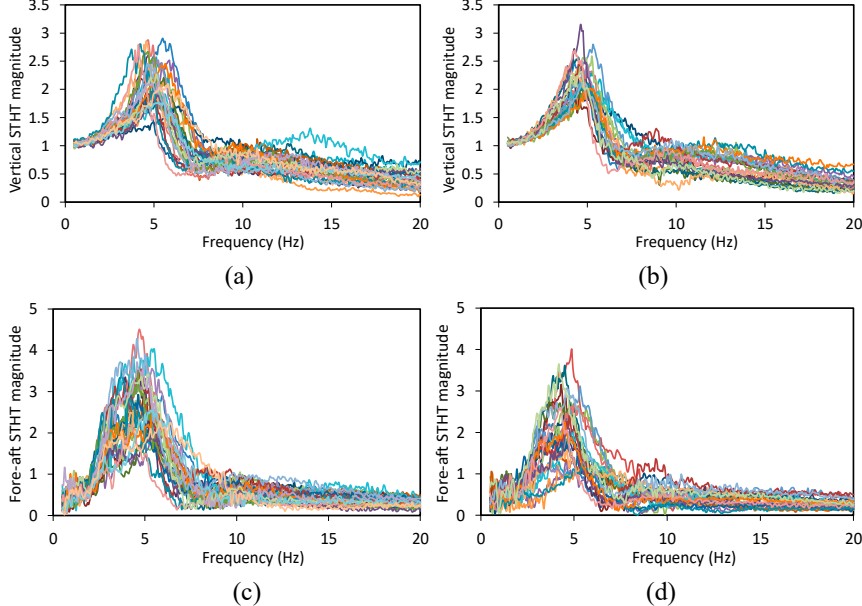

**Figure 3.** Comparisons of vertical and fore-aft STHT responses of 31 males and 27 females subjects: (**a**) male subjects—vertical STHT magnitude; and (**b**) female subjects—vertical STHT magnitude; (**c**) male subjects—fore-aft STHT; and (**d**) female subjects—fore-aft STHT (without back support; 0.50 m/s$^2$ RMS excitation).

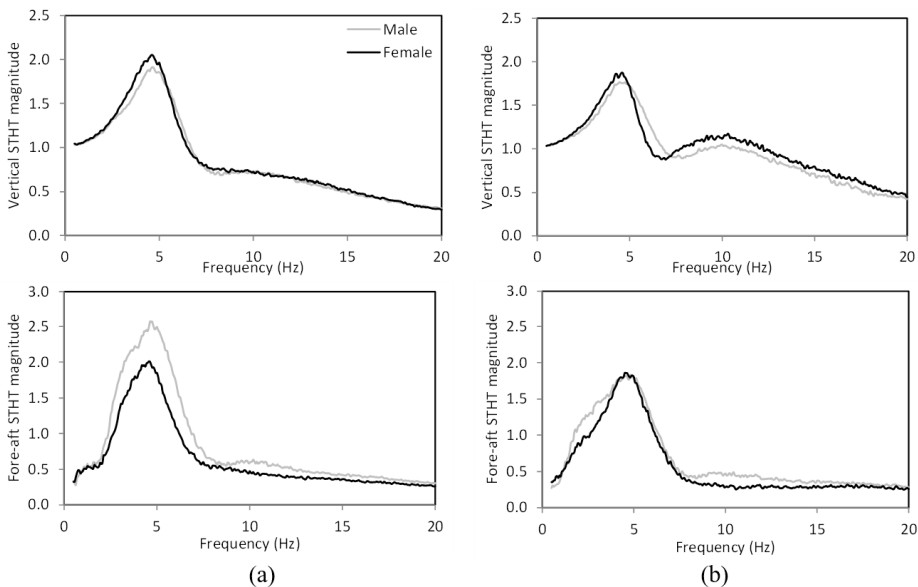

**Figure 4.** Mean vertical and fore-aft STHT magnitude responses of 31 males and 27 females: (**a**) without back support; and (**b**) with back support (excitation: 0.50 m/s$^2$).

Table 3 summarizes the means of peak vertical and fore-aft STHT magnitudes and the corresponding frequencies for the male and female subjects. The mean values are presented for the two back support conditions and three excitation magnitudes. With the exception of the 0.25 m/s$^2$ excitation, the mean vertical STHT magnitudes for both genders are generally comparable, while the primary resonance frequencies of males are considerably higher than those of females, irrespective of the sitting and excitation conditions. The fore-aft responses suggest a coupled effect with the sitting condition. The male subjects' responses exhibit considerably greater fore-aft peak magnitudes for the NB condition, while the peak magnitudes for both genders are comparable for the WB condition. The fore-aft responses also exhibit consistently greater primary resonance frequencies for the male subjects than those of the female subjects, irrespective of the sitting and excitation conditions.

**Table 3.** Means (standard deviations) of peak STHT magnitudes and the corresponding frequencies of the 31 male and 27 female subjects for the two sitting conditions and three levels of excitation.

| Excitation (m/s$^2$) | Male | | Female | |
|---|---|---|---|---|
| | NB | WB | NB | WB |
| Peak vertical STHT | | | | |
| 0.25 | 2.45 (0.41) | 2.17 (0.29) | 2.33 (0.27) | 2.06 (0.20) |
| 0.50 | 2.31 (0.35) | 2.07 (0.31) | 2.32 (0.29) | 2.01 (0.22) |
| 0.75 | 2.31 (0.34) | 2.06 (0.30) | 2.36 (0.32) | 2.09 (0.24) |
| Frequency corresponding to peak vertical STHT | | | | |
| 0.25 | 5.60 (0.57) | 5.48 (0.58) | 5.08 (0.53) | 4.79 (0.33) |
| 0.50 | 5.08 (0.54) | 4.99 (0.52) | 4.78 (0.37) | 4.48 (0.34) |
| 0.75 | 4.76 (0.48) | 4.65 (0.42) | 4.60 (0.38) | 4.36 (0.27) |
| Peak fore-aft STHT | | | | |
| 0.25 | 3.14 (0.64) | 2.35 (0.45) | 2.57 (0.72) | 2.35 (0.48) |
| 0.50 | 2.95 (0.70) | 2.17 (0.41) | 2.35 (0.75) | 2.18 (0.46) |
| 0.75 | 2.86 (0.80) | 2.16 (0.46) | 2.22 (0.73) | 2.07 (0.44) |
| Frequency corresponding to peak fore-aft STHT | | | | |
| 0.25 | 5.06 (0.75) | 5.05 (0.80) | 4.90 (0.54) | 4.87 (0.46) |
| 0.50 | 4.85 (0.60) | 4.62 (0.70) | 4.54 (0.55) | 4.52 (0.45) |
| 0.75 | 4.74 (0.57) | 4.50 (0.50) | 4.48 (0.49) | 4.34 (0.52) |

NB: without back support; WB: with back support.

Table 4 presents the results obtained from three-way ANOVA (gender, sitting condition, and excitation) of the peak STHT magnitudes and the primary resonance frequencies. The results show that the gender, sitting condition, and excitation levels have a significant ($p < 0.001$) difference in vertical mode primary resonance frequency, while the vertical peak magnitude is not significant between the genders and excitation levels. The interaction between gender and excitation magnitude is also significantly ($p < 0.05$) different on the vertical primary resonance frequency with an increase in the excitation level. Gender and excitation levels are also significantly different on the fore-aft primary resonance frequencies and peak magnitudes. Sitting condition further yields a significant difference in the fore-aft peak magnitude, while a significant difference is not evident in the fore-aft resonance frequency.

As an example, Figure 5a,b illustrate the back support effect on the mean vertical and fore-aft STHT magnitude responses of male and female subjects, respectively, under $0.50 \text{ m/s}^2$ excitation. These suggest important effects of back support, which are also evident in the results in Table 4. The back support effect on vertical STHT is particularly higher near the secondary resonance frequency compared with the primary resonance frequency. The fore-aft STHT responses exhibit a substantial back support effect for males, while the effect is lower for females.

**Table 4.** *p*–values obtained from a three-factor (G, BS, and E) analysis of variance (ANOVA) of the primary resonance frequency and peak STHT magnitude responses of the male and female subjects.

| Measure | G | BS | E | G × BS | G × E | BS × E | G × BS × E |
|---|---|---|---|---|---|---|---|
| Vertical resonance frequency | **<0.001** | **<0.001** | **<0.001** | 0.132 | **<0.05** | 0.968 | 0.976 |
| Vertical peak magnitude | 0.262 | **<0.001** | 0.168 | 0.694 | 0.181 | 0.980 | 0.913 |
| Fore-aft resonance frequency | **<0.05** | 0.489 | <0.001 | 0.171 | 0.741 | 0.116 | 0.886 |
| Fore-aft peak magnitude | **<0.001** | **<0.001** | **<0.01** | **<0.001** | 0.882 | 0.875 | 0.974 |

G—gender (male and female), BS—back support (without back support and with back support), E—excitation magnitude (0.25, 0.50, and 0.75 $\text{m/s}^2$), bold—$p < 0.05$.

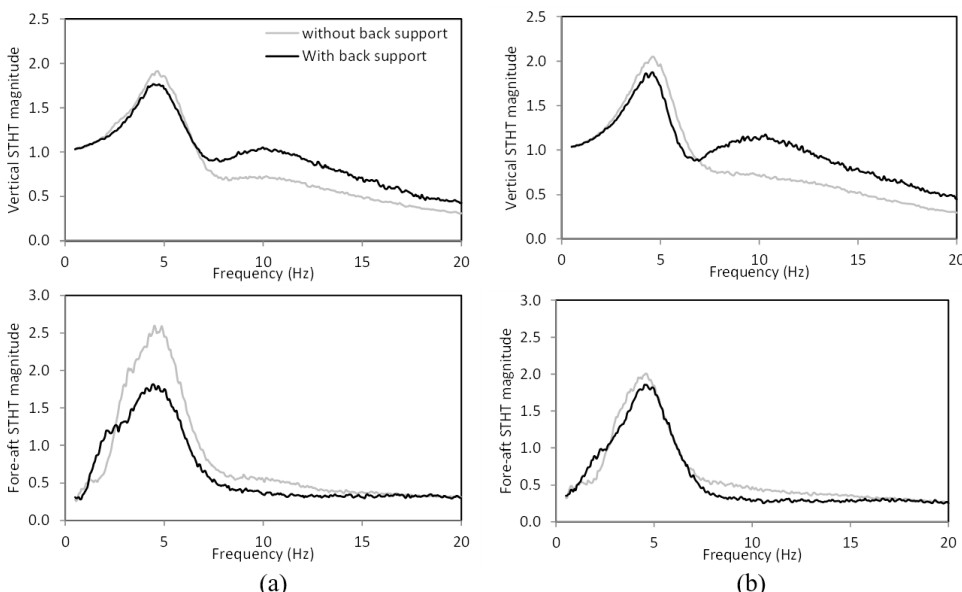

(a)　　　　　　　　　　　(b)

**Figure 5.** Comparisons of the mean STHT magnitudes of 31 male and 27 female subjects seated with and without back support under $0.50 \text{ m/s}^2$ excitations: (**a**) male subjects; and (**b**) female subjects.

Figure 6, as an example, illustrates the effect of excitation magnitude on mean magnitude responses of male and female subjects seated without a back support. The primary resonance frequency decreased with an increase in the excitation magnitude, as seem from the mean vertical STHT responses. This tendency a decrease in is evident with is as also

referred to as softening of the seated body with increasing excitation magnitude. Softening tendency is also evident in the fore-aft responses for male subjects, while the effect is minimal for female subjects. The peak vertical STHT magnitudes are generally comparable for both genders with an increase in excitation magnitude from 0.25 to 0.75 m/s$^2$, while the peak fore-aft magnitude decreased notably with increase in the excitation magnitude. Similar trends were also observed in the responses with the WB sitting condition. The change in the sitting condition from NB support to WB support caused 11–12% reductions in the mean values of peak vertical STHT magnitudes for both the gender groups, while the reductions in fore-aft STHT magnitude were up to 25% for all three excitations considered. The corresponding changes in primary resonance frequencies were approximately 0.11 and 0.28 Hz for the male and female subjects, respectively. The change in sitting condition from NB support to WB support caused the reduction in the peak fore-aft magnitudes up to 25% for the male subjects and only 8% for the female subjects, while the corresponding reductions in primary resonance frequencies were 0.16 and 0.06 Hz. The significant effects of back support on vertical and fore-aft STHT are also evident from the three-way ANOVA ($p < 0.001$), presented in Table 4. The results also show that the interaction of gender and back support condition is different on the peak fore-aft STHT magnitude ($p < 0.001$). Similar trends were also observed in the responses under other test conditions.

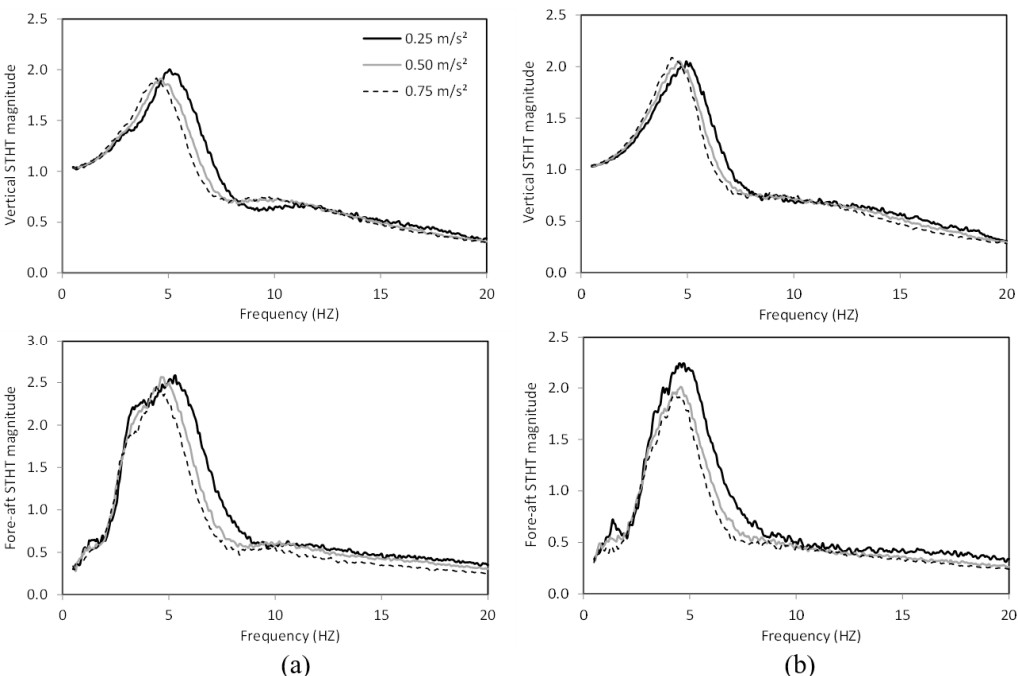

**Figure 6.** Influence of excitation level on mean STHT magnitude responses of subjects sitting without back support: (**a**) 31 males; and (**b**) 27 females.

### 3.2. Gender and Mass-Related Anthropometry

Figures 7 and 8 illustrate the effects of body mass on mean vertical and fore-aft STHT response magnitudes, respectively, for the male and female considering three mass ranges, listed in Table 2. The results are presented for both the back support conditions and 0.50 m/s$^2$ excitation. The vertical STHT responses of subjects within the three mass groups suggest that the effect of body mass is coupled with gender. The responses attained with male subjects show lower peak magnitude and lower primary resonance frequency with increasing body mass for both back support conditions. This effect is not clearly evident for females, especially for the NB condition. The fore-aft responses, however, do not show any clear trend with regard to body mass and gender, which may in part be caused by averaging the data.

The data acquired for individuals within each mass group are thus further analyzed to derive mean peak magnitudes and the corresponding frequencies for each mass group. The means and standard deviations of the peak vertical and fore-aft STHT magnitudes and corresponding frequencies are presented in Table 5 for both the genders and sitting conditions. Although the results are presented only for the 0.50 m/s$^2$ excitation, similar trends were observed for other excitations. The results show relatively higher primary resonance frequencies for the lower mass group subjects for both the genders and back support conditions, while a clear trend is generally not evident from the magnitude responses.

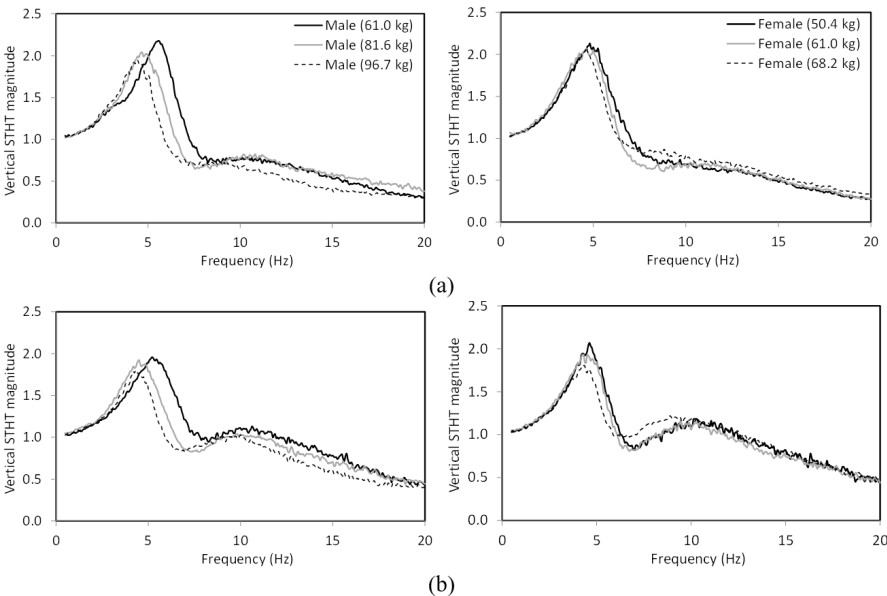

**Figure 7.** Mean vertical STHT magnitude responses of males and females within three different mass groups (**a**) without back support; and (**b**) with back support (excitation: 0.50 m/s$^2$).

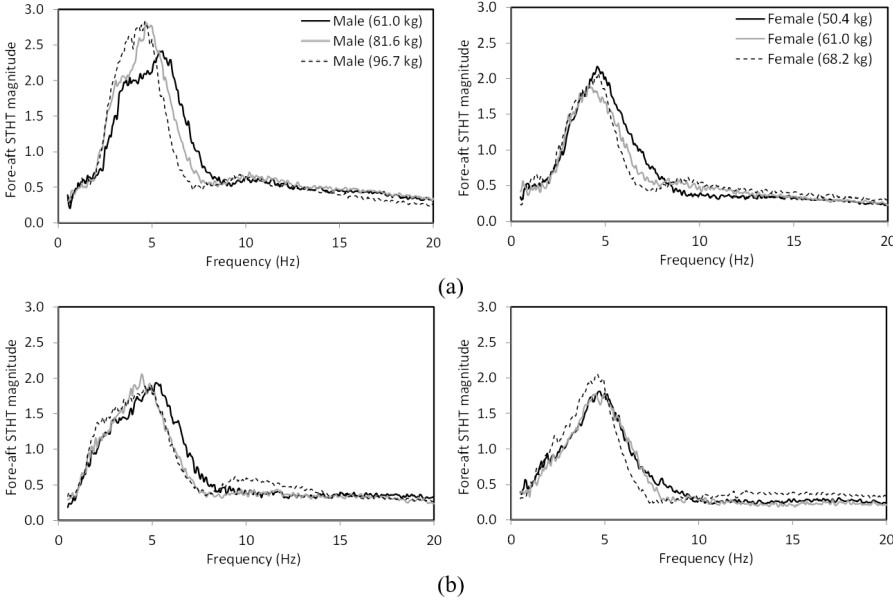

**Figure 8.** Mean fore-aft STHT magnitude responses of males and females within three different mass groups: (**a**) without back support; and (**b**) with back support (excitation: 0.50 m/s$^2$).

The effects of mass-related parameters, namely, BMI, body fat, and lean body mass, on mean vertical STHT responses of males and females are presented in Figure 9. The results, as an example, are presented for the NB sitting condition and 0.50 m/s$^2$ excitation, considering

three ranges of mass-related dimensions presented in Table 2. The results show that the primary resonance frequency decreases with an increase in all the mass-related parameters for the male subjects. The responses attained for the females, however, exhibit negligible effect of mass-related dimensions on the primary resonance frequency. The peak magnitudes for the three mass-related parameters are generally comparable for both genders.

**Table 5.** Means (standard deviations) of peak vertical and fore-aft STHT magnitudes and corresponding frequencies of male and female subjects within three ranges of body mass and two sitting conditions (0.50 m/s$^2$ excitation).

| Sitting Condition | Measures | Male | | | Female | | |
|---|---|---|---|---|---|---|---|
| | | 60.0 kg | 81.6 kg | 96.7 kg | 50.4 kg | 61.0 kg | 68.2 kg |
| NB | Vertical peak magnitude | 2.29 (0.31) | 2.31 (0.30) | 2.35 (0.39) | 2.29 (0.22) | 2.32 (0.25) | 2.27 (0.26) |
| | Vertical resonance frequency | 5.52 (0.36) | 4.82 (0.28) | 4.70 (0.35) | 4.98 (0.28) | 4.68 (0.47) | 4.67 (0.27) |
| WB | Vertical peak magnitude | 2.15 (0.22) | 2.13 (0.24) | 2.02 (0.40) | 2.08 (0.26) | 2.02 (0.23) | 1.89 (0.13) |
| | Vertical resonance frequency | 5.50 (0.40) | 4.77 (0.44) | 4.75 (0.39) | 4.71 (0.30) | 4.48 (0.16) | 4.43 (0.19) |
| NB | Fore-aft peak magnitude | 2.65 (0.57) | 3.02 (0.68) | 3.19 (0.74) | 2.40 (0.89) | 2.32 (0.82) | 2.32 (0.49) |
| | Fore-aft resonance frequency | 5.31 (0.57) | 4.63 (0.45) | 4.52 (0.43) | 4.59 (0.45) | 4.51 (0.70) | 4.47 (0.49) |
| WB | Fore-aft peak magnitude | 2.14 (0.47) | 2.25 (0.30) | 2.24 (0.39) | 2.08 (0.45) | 2.11 (0.50) | 2.34 (0.39) |
| | Fore-aft resonance frequency | 5.13 (0.50) | 4.49 (0.36) | 4.29 (0.82) | 4.72 (0.50) | 4.47 (0.39) | 4.42 (0.40) |

NB: without back support; WB: vertical back support.

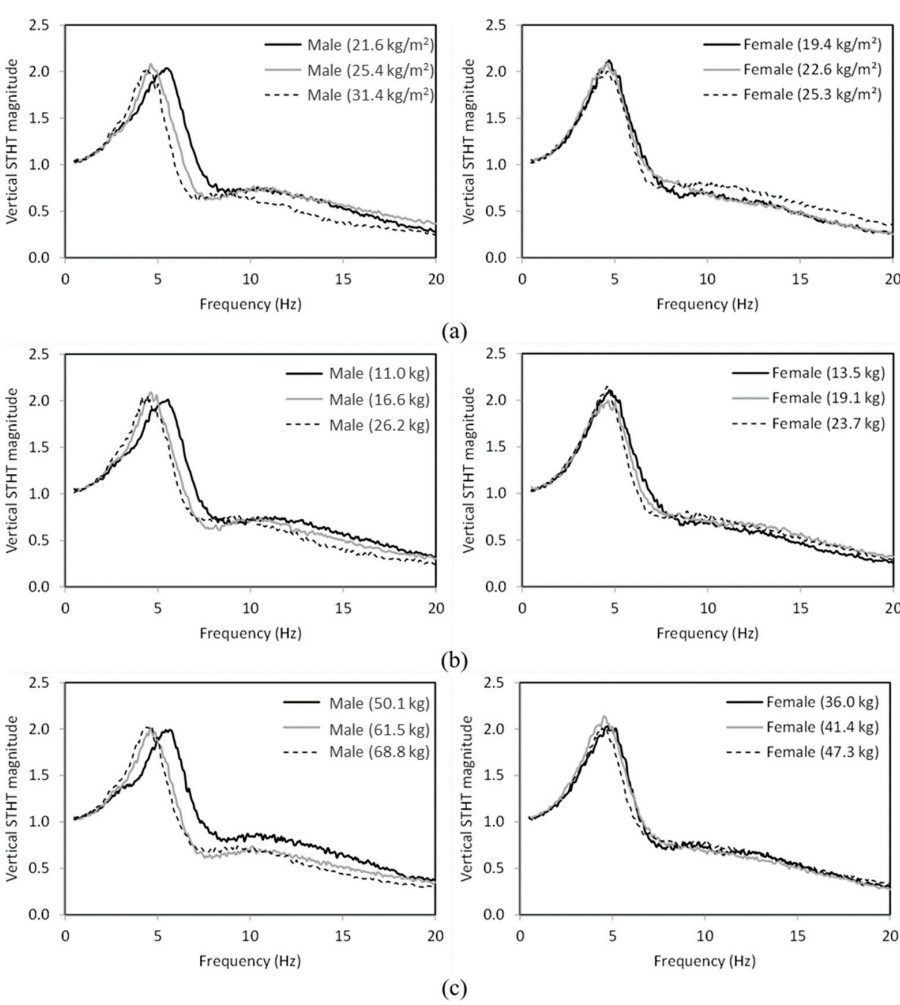

**Figure 9.** Effects of mass-related parameters on mean vertical STHT magnitude responses of males and females: (**a**) BMI; (**b**) body fat mass; and (**c**) lean body mass (NB support, 0.50 m/s$^2$ excitation).

Figure 10 illustrates the effects of same mass-related parameters on mean fore-aft STHT responses of the male and female subjects seated without a back support (NB) and exposed to 0.50 m/s² excitation. The results suggest similar notable effects of these parameters on the primary resonance frequencies for the males. The effects on primary resonance frequency are not evident for females, as it was observed in Figure 9. The female subjects' responses, however, exhibit notable effects of mass-related parameters on peak fore-aft magnitudes. The responses attained for male subjects generally show higher peak magnitudes for higher BMIs and lean body masses, while those attained for females show a different effect of BMI. The trends, however, are not very consistent, which is likely due to the averaging of the data.

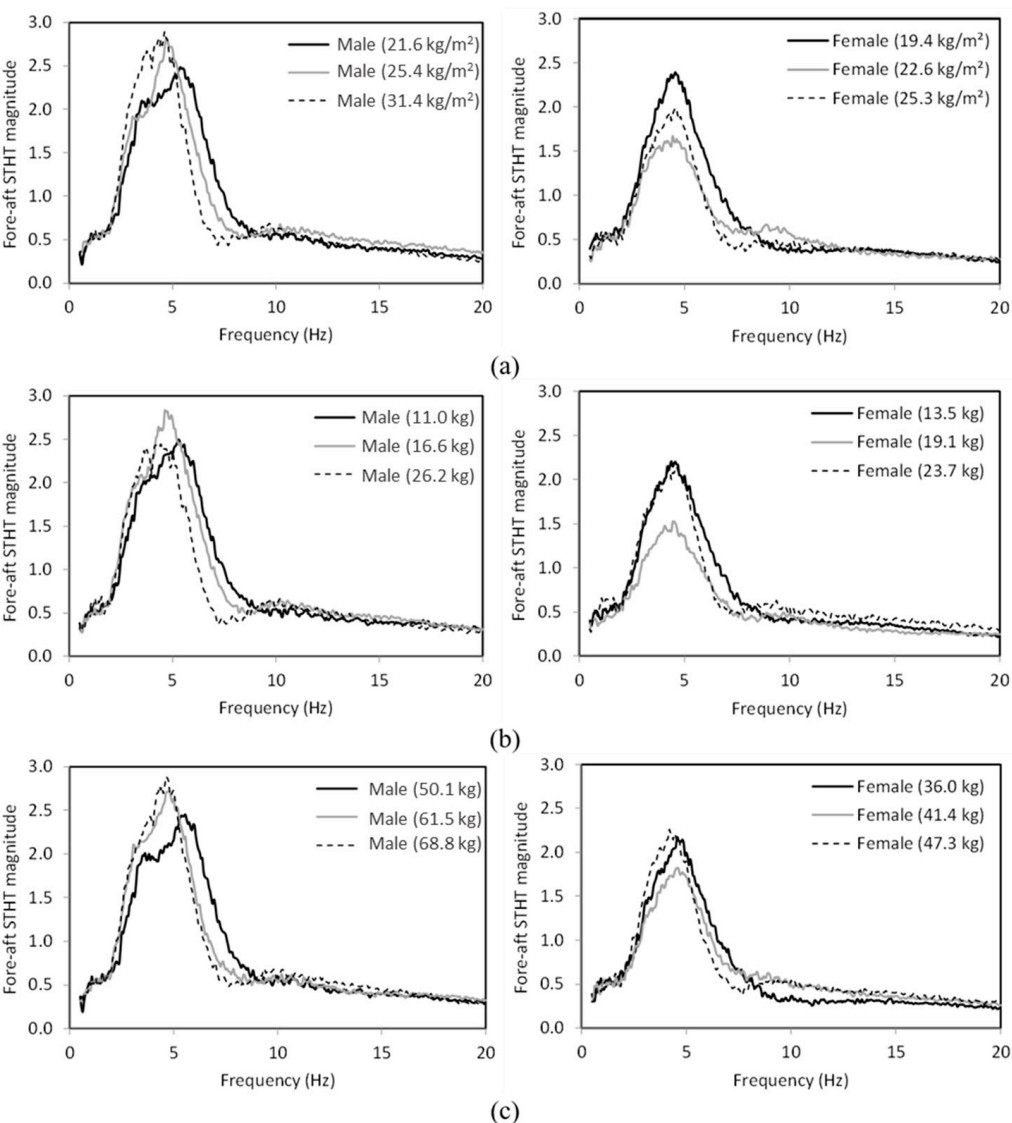

**Figure 10.** Effects of mass-related parameters on mean fore-aft STHT magnitude responses of male and female subjects: (**a**) BMI; (**b**) body fat mass; and (**c**) lean body mass (NB support, 0.50 m/s² excitation).

It should be noted that the mean body mass of male participants (79.8 kg) is substantially higher than that of female participants (60.1 kg), as shown in Table 1. The above results suggest that the gender effect is likely coupled with body mass. The data are thus further analyzed considering the responses obtained for males and females of comparable body mass and mass-related dimensions (BMI, body fat mass, and lean body mass), as listed in Table 2. Figure 11 compares the vertical and fore-aft STHT responses of male and female subjects within two comparable masses (60 and 70 kg), designated as G60 and

G70 in Table 2. The results are presented for both back support conditions and 0.50 m/s² excitation. The peak vertical and fore-aft STHT magnitudes for the males and females with comparable body masses are observed to be comparable, irrespective of the back support condition, while the primary resonance frequency observed for the females is lower than that of the males.

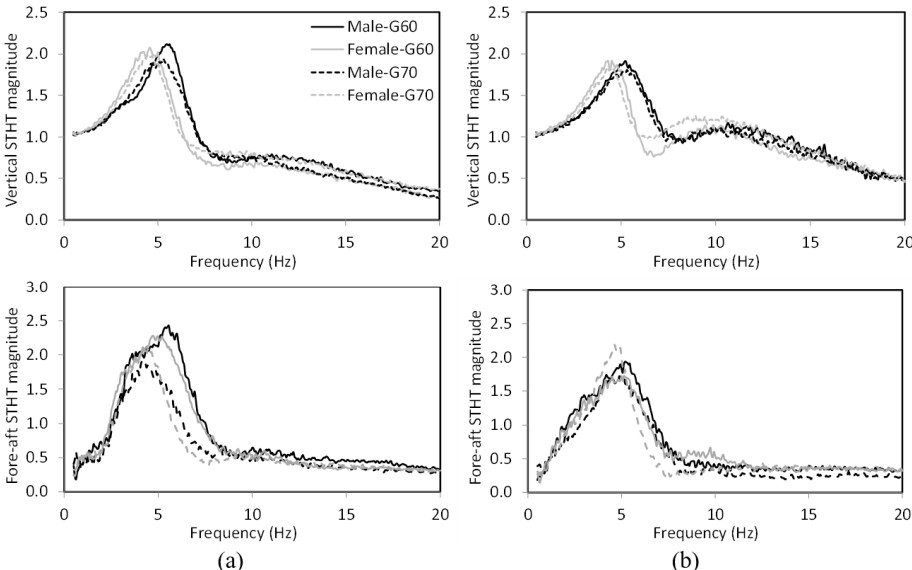

**Figure 11.** Mean STHT magnitude responses of male and female subjects within two mass groups (G60 and G70) for two different sitting conditions and 0.50 m/s² excitation: (**a**) without back support; and (**b**) with back support.

A paired *t*-test on the peak magnitude and corresponding frequencies for the G60 and G70 groups of the two genders was performed and the results are presented in Table 6. The results suggest that the primary resonance frequencies observed from the vertical STHT responses are generally significant between the male and female subjects. The responses to 0.75 m/s² RMS acceleration excitation, however, constitute an exception. The peak vertical magnitudes obtained for males and females, however, are not significantly different, irrespective of the sitting and excitation conditions, as it was observed in Figure 11. The peak fore-aft magnitudes and corresponding frequencies are also generally not significant between the two genders. From the results, it is deduced that the peak magnitude and the primary resonance frequency are coupled with the body mass, sitting condition, and excitation magnitude.

**Table 6.** *p*-values obtained from paired *t*-test of the primary resonance frequency and peak STHT magnitudes for the two gender groups of comparable body mass (60 and 70 kg) for the two sitting and three excitation conditions.

| Body Mass | Measure | Without Back Support | | | With Back Support | | |
|---|---|---|---|---|---|---|---|
| | | 0.25 m/s² | 0.50 m/s² | 0.75 m/s² | 0.25 m/s² | 0.50 m/s² | 0.75 m/s² |
| G60 | Vertical resonance frequency | <0.001 | <0.05 | 0.135 | <0.001 | <0.001 | <0.05 |
| | Vertical peak magnitude | 0.536 | 0.486 | 0.475 | 0.291 | 0.614 | 0.921 |
| G70 | Vertical resonance frequency | <0.05 | <0.05 | 0.123 | <0.05 | <0.01 | 0.120 |
| | Vertical peak magnitude | 0.846 | 0.605 | 0.886 | 0.132 | 0.407 | 0.968 |
| G60 | Fore-and-aft resonance frequency | 0.054 | 0.144 | 0.281 | <0.05 | 0.190 | 0.495 |
| | Fore-and-aft peak magnitude | 0.161 | 0.454 | 0.242 | 0.176 | 0.749 | 0.528 |
| G70 | Fore-and-aft resonance frequency | 0.175 | 0.153 | 0.198 | 0.831 | 0.595 | 0.607 |
| | Fore-and-aft peak magnitude | 0.892 | 0.326 | 0.691 | <0.05 | 0.200 | 0.979 |

The mean vertical and fore-aft STHT responses of the males and females with comparable BMI (25.4 kg/m$^2$ for males and 25.3 kg/m$^2$ for females), body fat mass (26.2 kg for males and 23.7 kg for females), and lean body mass (50.1 kg for males and 47.3 kg for females), listed in Table 2, are also compared in Figure 12 for the NB sitting condition and 0.50 m/s$^2$ excitation. The results suggest a strong gender effect on the magnitude of the STHT response, particularly, for the comparable body fat mass and lean body mass. The effects are very pronounced in the vicinity of the peak magnitude and primary resonance frequency for the vertical as well as fore-aft STHT responses. The gender effect, however, is very small for comparable BMI when the vertical STHT responses are considered. The female subjects' response revealed a considerably higher peak vertical STHT magnitude compared with that of the male subjects with comparable body fat mass, although the corresponding frequencies are similar for both genders. The vertical and fore-aft responses of females revealed considerably lower primary resonance frequency when compared to that observed for males with comparable lean body mass, as seen in Figure 12c. The male subjects' responses, however, show notably higher peak fore-aft STHT magnitudes than the female subjects with comparable BMI, body fat mass, and lean body mass. Similarly, a substantially higher primary resonance frequency is observed for the male subjects than the female subjects with comparable lean body mass.

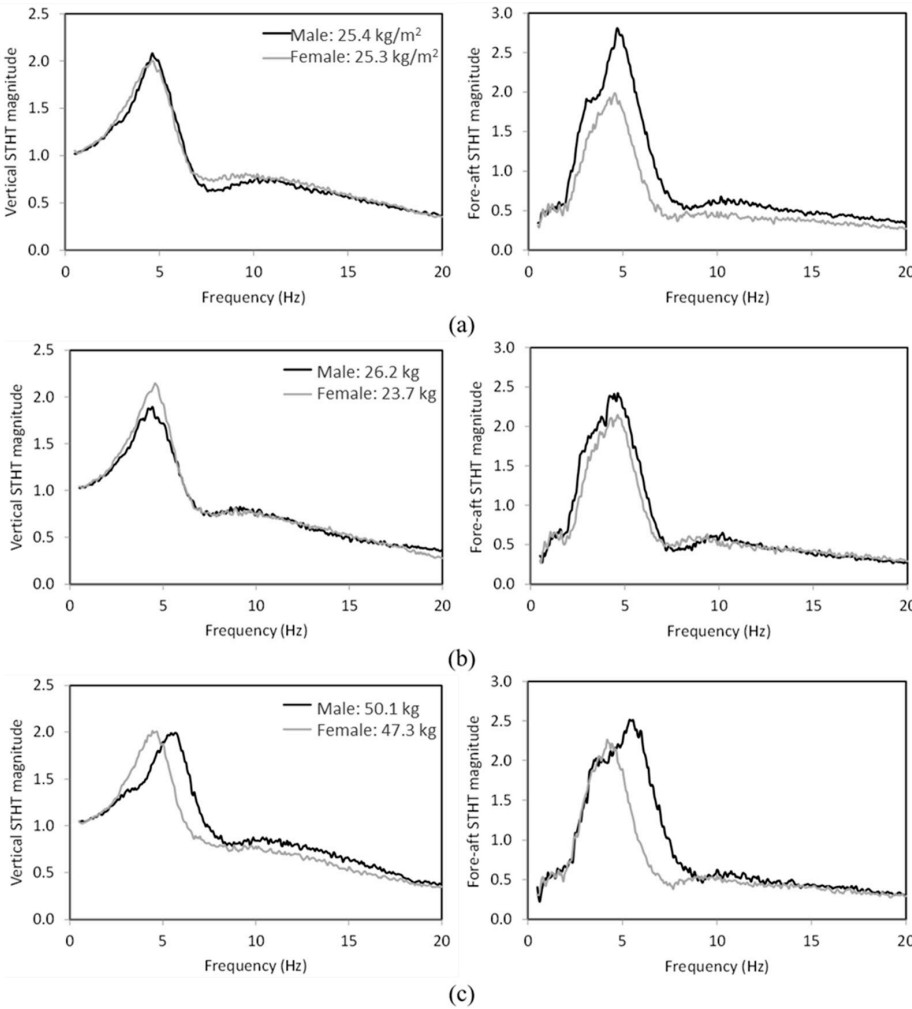

**Figure 12.** Effects of gender on mean STHT magnitudes considering comparable mass-related parameters of males and females: (**a**) BMI; (**b**) fat body mass; and (**c**) lean body mass (NB support, 0.50 m/s$^2$ excitation).

Results obtained from paired *t*-tests of selected datasets of males and females with comparable ranges of lean body mass show significant differences ($p < 0.05$) in primary resonance frequency observed from the vertical STHT responses, irrespective of the back support and excitation conditions (Table 7). All other mass-related parameters, however, have no significant difference in the vertical peak magnitudes of STHT responses, irrespective of the sitting and excitation conditions.

**Table 7.** *p*-values obtained from independent *t*-tests of peak vertical STHT magnitude and primary resonance frequency between the male and female subjects of comparable anthropometric dimensions for the two back support conditions and three excitation magnitudes.

| Anthropometric Parameters | | Primary Resonance Frequency | | | | | | Peak Magnitude | | | | | |
|---|---|---|---|---|---|---|---|---|---|---|---|---|---|
| | | Without Back Support | | | With Back Support | | | Without Back Support | | | With Back Support | | |
| | | 0.25 m/s$^2$ | 0.50 m/s$^2$ | 0.75 m/s$^2$ | 0.25 m/s$^2$ | 0.50 m/s$^2$ | 0.75 m/s$^2$ | 0.25 m/s$^2$ | 0.50 m/s$^2$ | 0.75 m/s$^2$ | 0.25 m/s$^2$ | 0.50 m/s$^2$ | 0.75 m/s$^2$ |
| Mass-related | BMI | 0.118 | 0.467 | 0.370 | <0.05 | <0.01 | 0.201 | 0.887 | 0.369 | 0.450 | 0.703 | 0.084 | 0.784 |
| | Body fat mass | 0.162 | 0.430 | 0.684 | <0.01 | <0.05 | 0.110 | 0.682 | 0.627 | 0.439 | 0.664 | 0.218 | 0.656 |
| | Percent body fat | 0.188 | 0.992 | 0.490 | <0.05 | 0.084 | 0.136 | 0.068 | 0.193 | 0.060 | 0.354 | 0.497 | 0.521 |
| | Lean body mass | <0.01 | <0.01 | <0.01 | <0.01 | <0.01 | <0.05 | 0.356 | 0.927 | 0.631 | 0.192 | 0.057 | 0.800 |
| Build-related | Hip circumference | 0.365 | 0.108 | 0.704 | <0.05 | 0.094 | 0.212 | 0.127 | 0.750 | 0.584 | 0.472 | 0.135 | 0.240 |
| | Seat–pan contact area | 0.327 | 0.866 | 0.779 | <0.01 | 0.062 | 0.240 | 0.195 | 0.260 | 0.347 | 0.135 | 0.085 | 0.053 |
| | Mean contact pressure | 0.260 | 0.055 | 0.995 | <0.05 | <0.05 | 0.089 | 0.395 | 0.348 | 0.738 | 0.286 | 0.093 | 0.190 |
| Stature-related | Stature | <0.01 | <0.05 | <0.01 | <0.01 | <0.01 | <0.01 | 0.553 | 0.058 | 0.182 | 0.458 | 0.805 | 0.104 |
| | Sitting height | <0.05 | 0.133 | <0.05 | <0.01 | <0.01 | 0.414 | 0.145 | <0.01 | <0.05 | 0.895 | 0.731 | 0.062 |
| | C7 height | 0.097 | 0.650 | 0.205 | <0.05 | <0.05 | 0.401 | <0.05 | 0.262 | <0.05 | 0.771 | 0.627 | 0.408 |

### 3.3. Gender and Build-Related Anthropometry

Figure 13, as an example, presents the effects of hip circumference on mean vertical STHT responses of males and females seated without a back support and subject to 0.50 m/s$^2$ excitation. The results are obtained by grouping the datasets of male and female subjects with hip circumference within three ranges, whose mean values are listed in Table 2. Since the hip circumference directly affects the effective body–seat contact area and mean contact pressure, these data reported in [14] were also analyzed for the same subject groups. Table 2 summarizes the ranges of mean effective area and contact pressure for the two subject groups. Lower hip circumference and contact area yield relatively higher primary resonance frequencies for both the gender groups. A similar trend is also evident with the mean contact pressure for the male subjects; however, the responses are comparable for most of the frequency range with the female subjects. The peak STHT magnitudes of the male subjects are lower for the higher dimensions of the hip circumference and contact area, while no trend is evident for mean contact pressure. For the female subjects, the peak magnitude is comparable for three ranges of hip circumference, while the peak magnitude is lower for the higher dimension of the contact area. No trend in the peak magnitude is evident for the female subjects on the mean contact pressure as in the case of male subjects. Similar trends were also observed for the frequencies corresponding to peak fore-aft STHT response with regard to the hip circumference, effective contact area, and mean contact pressure, while no clear trend was observed for the peak magnitude (results not shown).

The results in Figures 11 and 12 suggest that the gender effect on the STHT responses could be effectively evaluated by considering comparable mass-related parameters of the two gender groups. A similar approach is also undertaken to evaluate the gender effect by grouping responses of male and female subjects with comparable hip circumference (male: 102.8 cm and female: 100.4 cm), contact area (male: 666 cm$^2$ and female: 682 cm$^2$), and mean contact pressure (male: 9.1 N/cm$^2$ and female: 9.3 N/cm$^2$). The results obtained for the NB condition and 0.50 m/s$^2$ excitation are presented in Figure 14, as an example. The results show that the peak vertical magnitudes of female subjects are slightly higher than those of the male subjects for all the build-related parameters considered. The peak fore-aft magnitudes, however, show an opposing trend in gender effect, i.e., the male

subjects exhibit higher peak magnitudes than the female subjects for all three build-related parameters. Moreover, the primary resonance frequencies for the female subjects are generally relatively lower than those for the male subjects with comparable build-related parameters. The results obtained from the paired *t*-tests of the selected datasets show that the peak vertical STHT magnitudes and the primary resonance frequencies of male and female subjects are not generally significant when comparable build-related parameters are considered, irrespective of the back support and excitation conditions (Table 7).

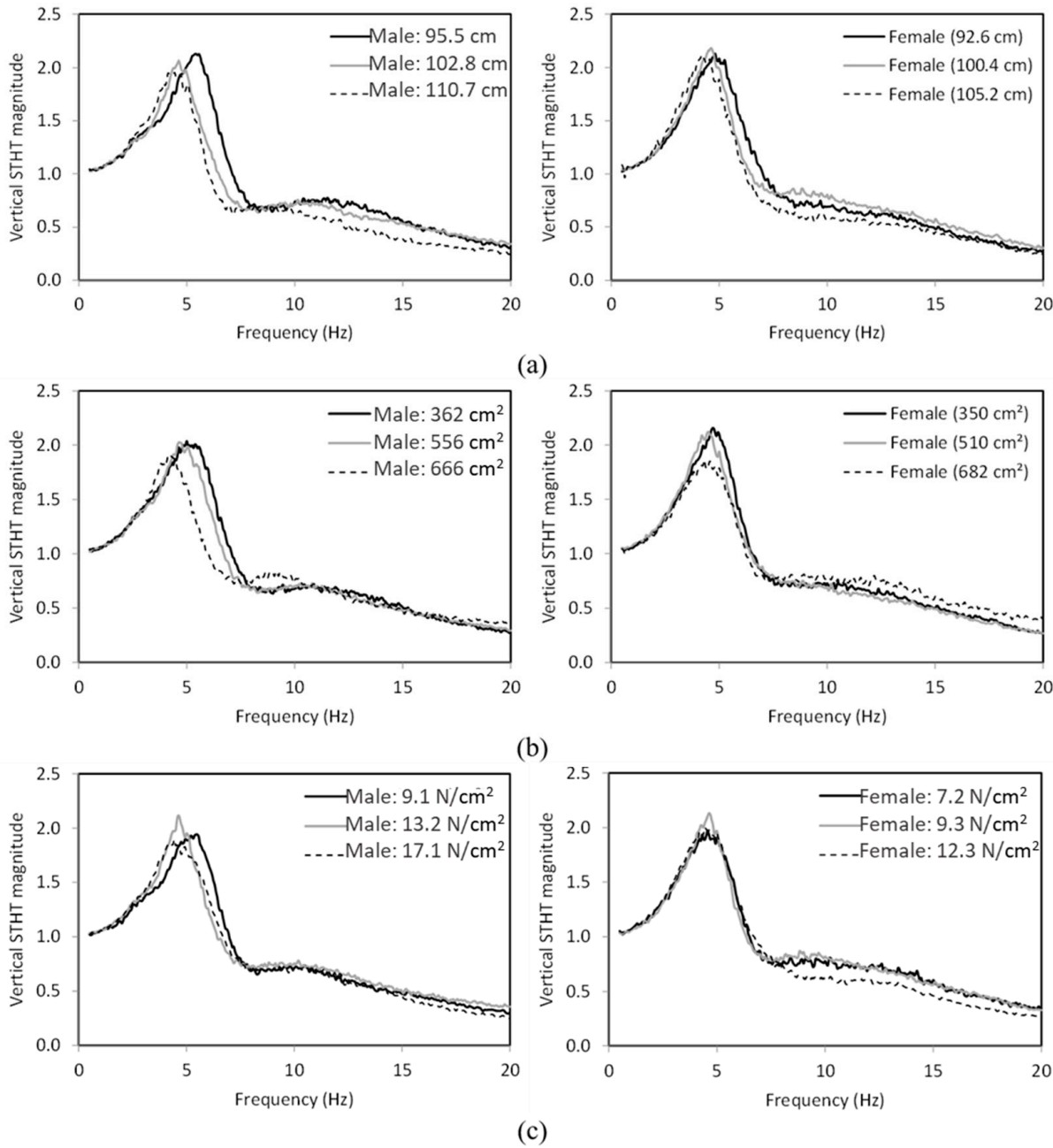

**Figure 13.** Influences of hip circumference, effective body-seat contact area, and mean contact pressure on mean vertical STHT magnitudes of males and females: (**a**) hip circumference; (**b**) seat–pan contact area; and (**c**) mean contact pressure (NB support, 0.50 m/s$^2$ excitation).

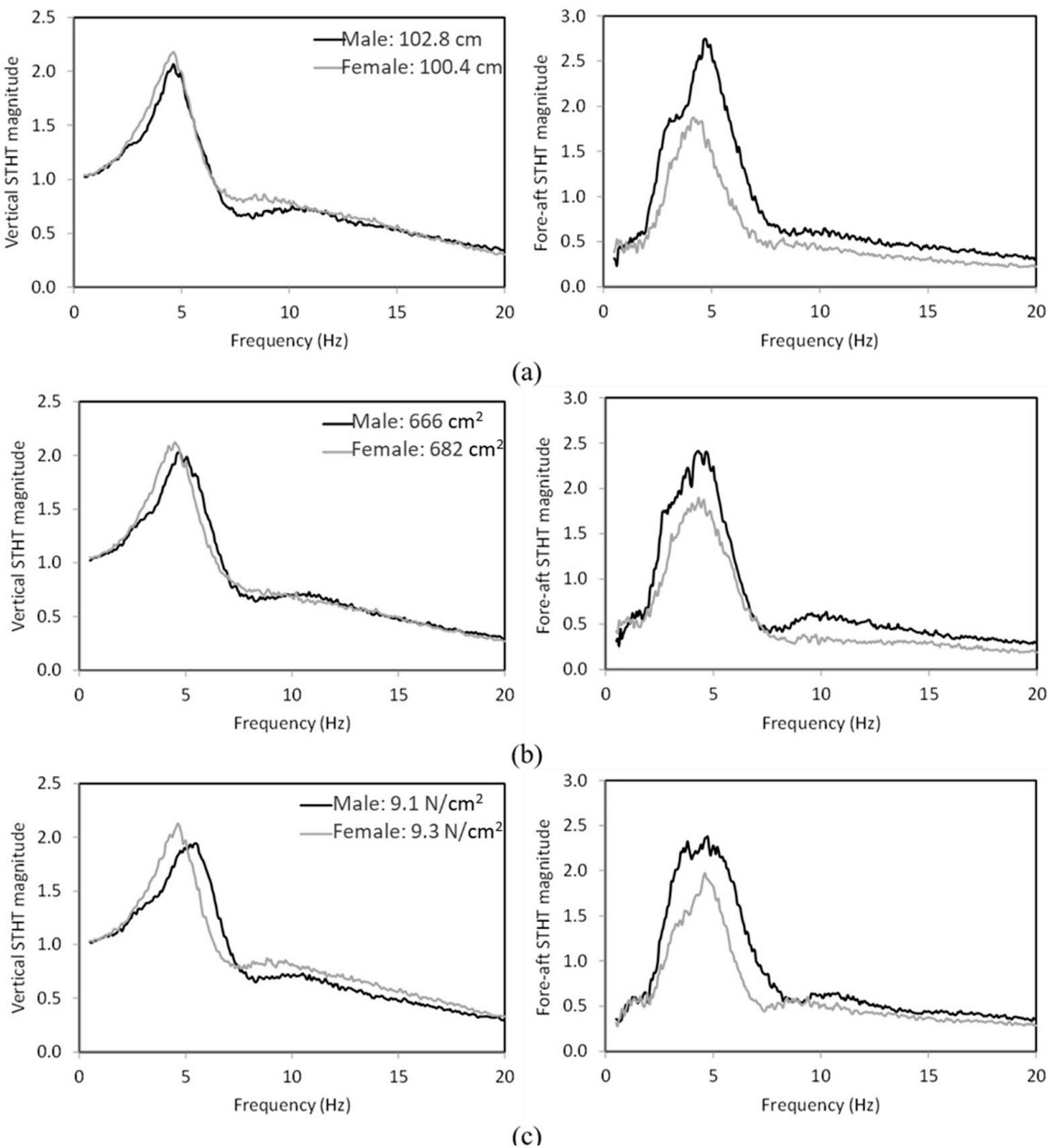

**Figure 14.** Effect of gender on mean STHT vertical and fore-aft magnitude responses considering comparable build-related parameters: (**a**) hip circumference; (**b**) seat–pan contact area; and (**c**) mean contact pressure (NB support, 0.50 m/s$^2$ excitation).

*3.4. Gender and Stature-Related Anthropometry*

Figures 15 and 16 present the influences of stature-related parameters on mean vertical and fore-aft STHT responses, respectively, obtained for both the gender groups considering NB sitting condition and 0.50 m/s$^2$ excitation. The results are obtained for three ranges of build-related parameters, namely, stature, sitting height and C7 height, presented in Table 2. The results suggest relatively small influences of chosen parameters on the vertical STHT responses. The effects on the fore-aft responses, however, are relatively more significant, particularly for the female subjects. Furthermore, no particular trend on the peak magnitude as well as primary resonance frequency is evident.

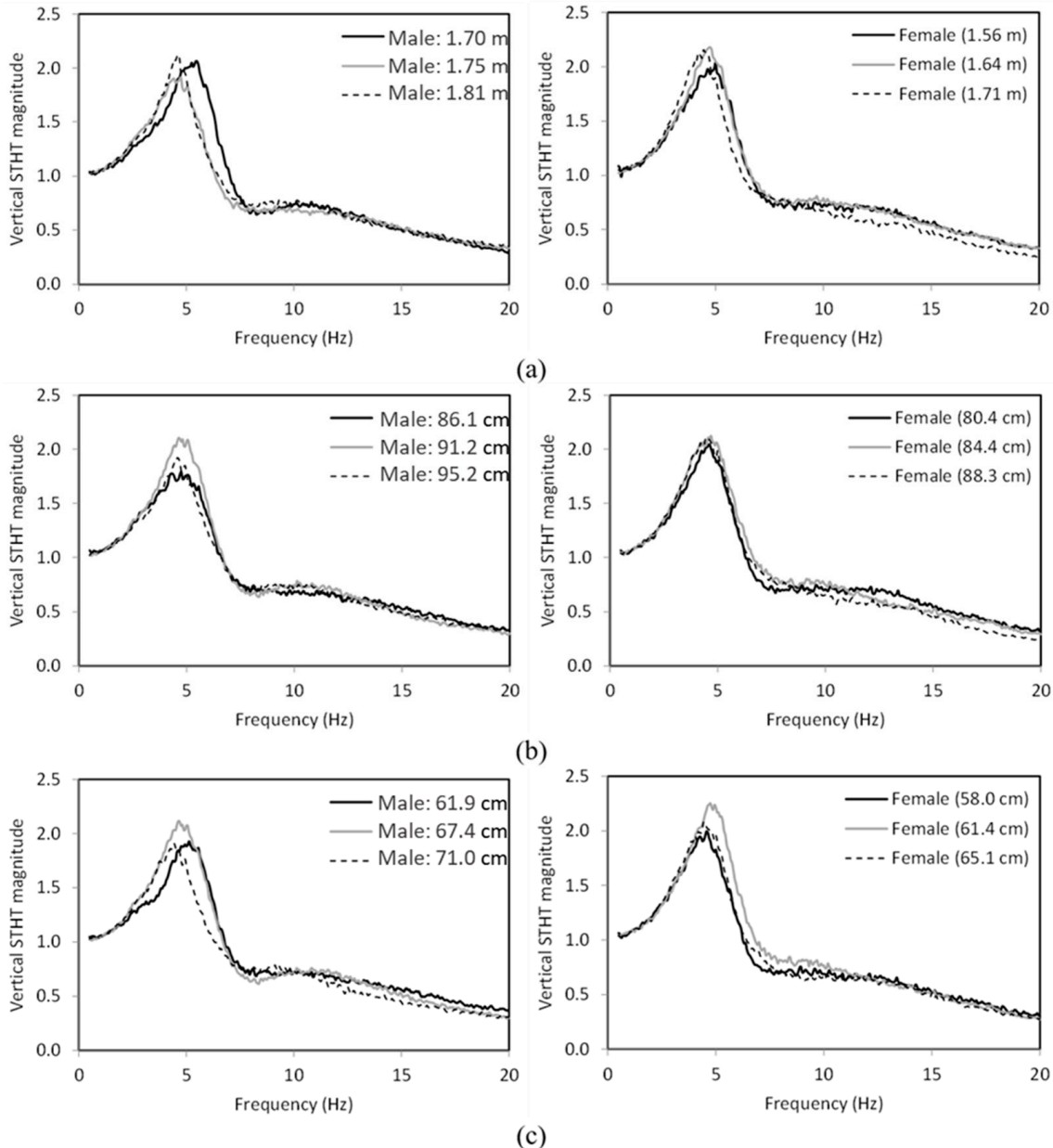

**Figure 15.** Effects of stature-related parameters on mean vertical STHT magnitude responses of male and female subjects: (**a**) stature; (**b**) sitting height; and (**c**) C7 height (NB support, 0.50 m/s$^2$ excitation).

The gender effect is further evaluated considering selected datasets of female subjects of comparable stature-related anthropometric parameters, as seen in Figure 17, for the NB condition and 0.50 m/s$^2$ excitation. The ranges for the comparable values of the stature (male: 1.70 m; female: 1.64 m), sitting height (male: 86.1 cm; female: 84.4 cm), and C7 height (male: 61.9 cm; female: 61.4 cm) are listed in Table 2. The results show substantial differences between vertical and fore-aft STHT responses of males and females, when comparable stature-related parameters are considered for both the genders. The peak vertical magnitudes of the female subjects are higher than the male subjects, while the opposite trend is evident in the peak fore-aft magnitude. The results obtained for females generally show lower primary resonance frequencies compared to those observed for the males. This is further confirmed by the results attained from paired *t*-tests of selected datasets, which show that the primary resonance frequency of the vertical STHT responses of the male and female subjects are significantly different considering comparable stature, especially for the NB sitting condition (Table 7).

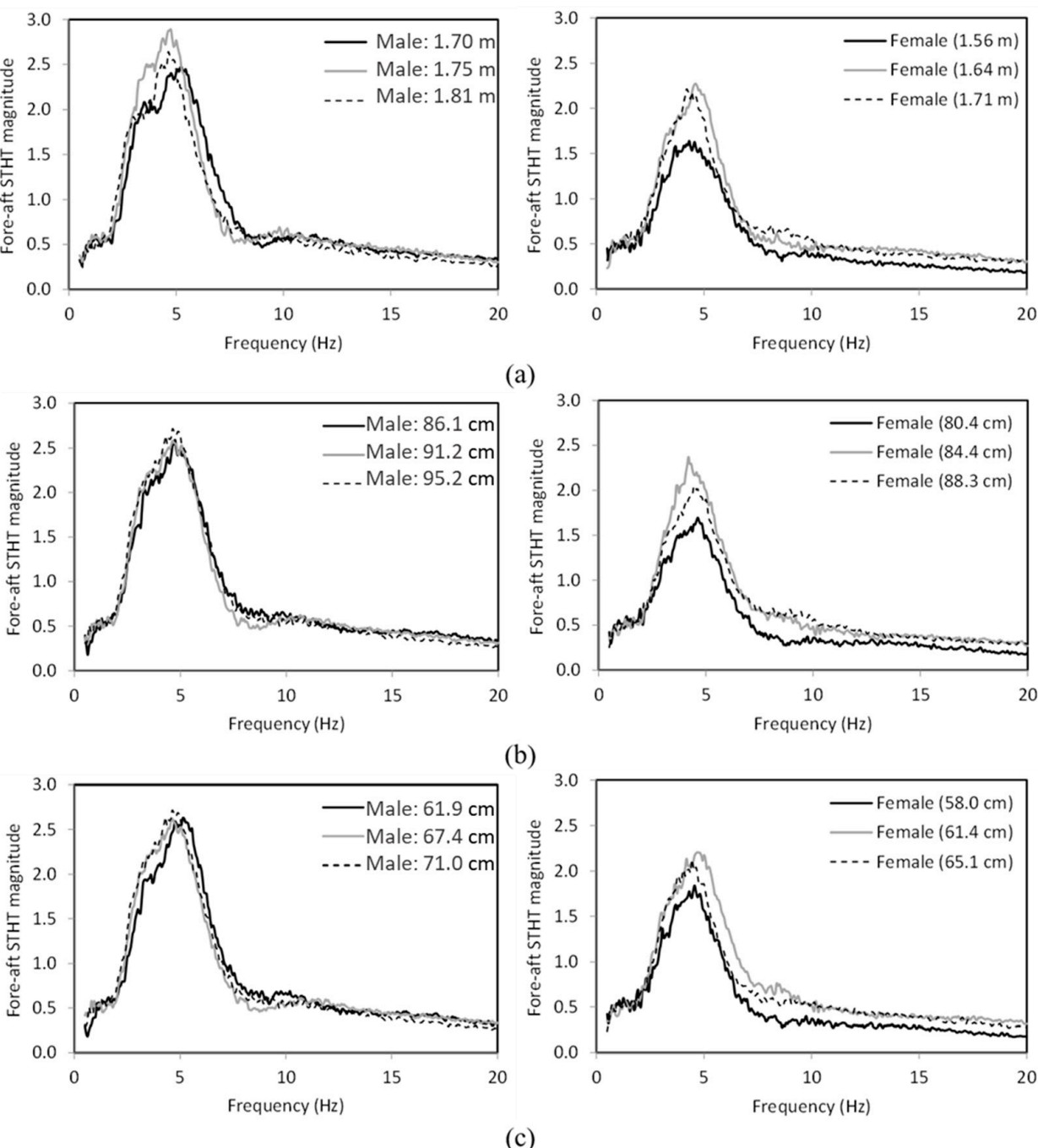

**Figure 16.** Effects of stature-related parameters on mean fore-aft STHT magnitude responses of male and female subjects: (**a**) stature; (**b**) sitting height; and (**c**) C7 height (NB support, 0.50 m/s$^2$ excitation).

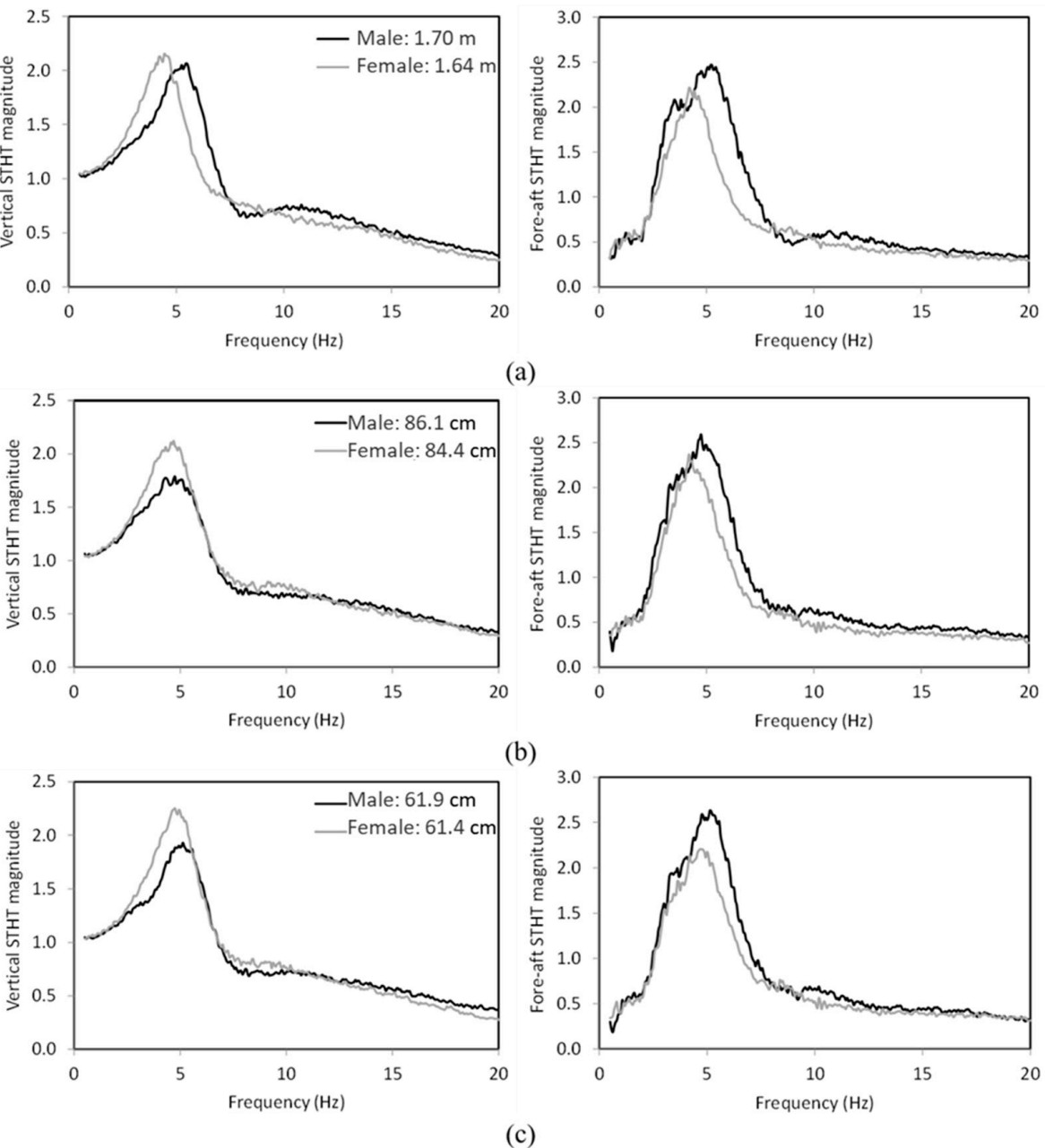

**Figure 17.** Effects of gender on mean vertical and fore-aft STHT magnitude responses considering comparable stature-related parameters of two gender groups: (**a**) stature; (**b**) sitting height; and (**c**) C7 height (NB support, 0.50 m/s$^2$ excitation).

## 4. Discussions

### 4.1. STHT Response Characteristics and Gender Effect

Biodynamic response characteristics measured in terms of vertical and fore-aft STHT responses of the human body seated on an elastic seat and exposed to vertical WBV show a strong gender effect (Tables 3 and 4), which is further coupled with the sitting and excitation conditions. The gender effect on the STHT responses have investigated only in few studies [3]. Only a few studies are thus available for a comparison of the results of the present study with the published data. Furthermore, the vast majority of studies reporting STHT responses have considered the human body seated on a rigid seat. Owing to the notable effects of elastic seats on biodynamic responses [26], the results obtained in this study could not be compared with the reported data.

The present study shows that peak STHT magnitude occurs in a relatively broader frequency range for the male subjects (4.13–6.38 Hz and 3.50–6.00 Hz for vertical and fore-aft responses, respectively) as compared with the female subjects (4.00–5.31 Hz and 3.50–5.69 Hz for vertical and fore-aft responses, respectively) (Figure 3). The present study further shows different degrees of scattering in the data in the vicinity of the primary resonance frequency for the male subjects (16%–38% and 28%–38% for vertical and fore-aft STHT responses, respectively) and female subjects (14%–20% and 35%–41% for vertical and fore-aft STHT responses, respectively) (Figure 3). Wilder et al. [27] reported widely scatter data of transmissibility for the female subjects; however, the dimensions of the subjects were not provided. Large variability in the primary resonance frequency with the male subjects in the present study may probably be due to large variation in the anthropometric dimensions of the male subjects (body mass: 55–106 kg, hip circumference: 88–116 cm, stature: 1.59–1.92 m) as compared with the female subjects (body mass: 45.5–72.5 kg, hip circumference: 89.5–109 cm, stature: 1.48–1.73 m). Dewangan et al. [14] also opined that large variation can be found in the primary resonance frequency of the subjects seated on the rigid seat. A relatively larger scatter of the female subjects as compared with the male subjects for the fore-aft STHT responses shows the effect of other contributory factors. Wilder et al. [27] and Lundström et al. [28] suggested breast mass, breast support, and oscillations could partly alter the responses of the female subjects. Lundström et al. [28] also investigated the breast supports effect of on energy absorption of the seated body under vertical vibration, although the contributions of the breast support was not shown. The results in the present study show that the vertical back support reduces variability of the data when compared to the NB condition for both the genders. A similar trend in the scatter has also been reported in previous studies [14,24,25]. Dewangan et al. [14] reported relatively higher vertical STHT magnitude near secondary resonance for the female subjects than the male subjects, while seated on a rigid seat. This trend, however, is not visible in the present study (Figure 3). The mean responses of the male and female subjects near the secondary resonance are comparable for the NB sitting condition. Relatively higher magnitude at secondary resonance for the female subjects than the male subjects with the WB sitting condition (Figure 4) may be due to the coupling effect of gender with the sitting condition.

The present study shows a relatively higher peak vertical STHT magnitude for the female subjects than the male subjects (Figure 4). The means of the peak vertical magnitudes obtained for all the subjects (Table 3), however, are comparable for both genders, irrespective of back support and excitation conditions. This difference in the results is likely due to the data averaging. Since the peak STHT magnitude for the male subjects was obtained in a wide range of frequencies, the mean peak STHT response was relatively lower. The present study shows significantly higher ($p < 0.001$) primary resonance frequency for the male subjects as compared with the female subjects, irrespective of back support and excitation conditions (Tables 3 and 4). Wilder et al. [27] and Dewangan et al. [14] also reported relatively higher primary resonance frequencies of the male subjects than those obtained with the female subjects. However, the peak vertical magnitude data of the two genders are opposite to that reported by Dewangan et al. [14]. The median vertical STHT response reported by Griffin et al. [29] considering a rigid seat exhibits higher peak magnitudes from male subjects as compared with the female subjects. The results in the present study show higher mean transmissibility of males than the females in the 1.25–4 Hz range. An opposite trend, however, was observed at frequencies above 5 Hz. Griffin and Whitham [30] measured the gender difference on the vertical STHT magnitude measured with a rigid seat at two discrete frequencies (4 and 16 Hz) and showed higher STHT magnitudes for the female subjects at both frequencies. The differences in the STHT responses among different studies could also be due to the measurement system, anthropometric dimensions of the subjects, and seat type. Dewangan et al. [14] suggested the coupling of the anthropometric data with the STHT responses.

The morphology of the male and female subjects is characteristically different, and thus the two genders are anatomically different. The wider pelvic size in a female [31] may be attributed to lower lumbopelvic stability among females [32]. The curvature and tilt of the lumbar spine are also different between the two genders. Therefore, the pelvis orientation and pelvis rotation are different when the male and female subjects occupy the seat. A female tends to demonstrate greater anterior pelvic tilt compared with a male [33,34]. The stiffness of the lumbar spine together with the intra-abdominal pressure in the abdominal cavity preserve stability of the spine and [35]. The transmission of seat vibration through the body may thus depend on many contributory factors related to anthropometric dimensions. Studies on the gender effect on the STHT responses, however, have generally not considered the possible coupling of various anthropometric dimensions [27,29,30]. Kitazaki and Griffin [36] suggested that the resonance of the body may be attributed to axial and shear deformations of tissues beneath the pelvis. the majority of the body fat in a female lies within the pelvis and thighs. Owing to the relatively higher tissue mass on the seat pan, the axial and shear deformation in a female may thus be different from a male. Possibly greater pelvis motion with the females is attributed to higher peak vertical magnitude. Therefore, the comparison of the STHT responses of comparable body dimensions may provide more insight into the gender effect. The comparison of the peak STHT magnitudes and corresponding frequencies of the present study with those in Dewangan et al.'s work [14] shows relatively higher peak vertical STHT magnitudes and lower primary resonance frequencies in the present study for both the genders, irrespective of sitting and excitation conditions. Furthermore, the peak fore-aft STHT magnitude is lower in this study for the NB sitting condition; however, the opposite trend was obtained for the WB condition for both genders. The comparison of the fore-aft primary resonance frequency revealed that the resonance frequency was lower for the NB sitting condition, while the primary resonance frequency was generally lower for the WB sitting condition for both genders. The variation between the two studies may most likely be due to the coupling effect of the body with the seat.

Changing the sitting condition from NB to WB significantly reduced the peak vertical and fore-aft STHT magnitude ($p < 0.001$), while the effect is different for the male and female subjects (Tables 3 and 4). The effect of back support obtained in the present study is in line with the previous studies [8,12,14,24,25]. Sitting support changes pelvic rotation and spine orientation. Sitting without a back support causes the lumber spine to assume lordotic posture (inward curvature of the lumbar spine) in order to obtain balanced sitting [37]. The lumbar lordosis increases with back support and thus the derotation of the pelvis occurs [38]. A backrest also serves as a constraint for the motion of the spine in the sagittal plane, thus reducing the pitch motion of the torso and thus affecting the fore-aft motion of the head. A backrest may also reduce axial deformation of the lower lumbar spine and thus influence the transmission of vibration to the head. The backrest changes the muscle tension of the torso and acts as an additional driving point. Sitting on the elastic seat slightly changes body–seat contact area and mean contact pressure (Table 3). Sitting on an elastic seat with a back support yields relatively lower mean contact pressure when compared to sitting without a back support [26]. The transmission of seat vibration to the head is thus also affected by the back support condition. The results in this study show notable differences in the responses obtained with males and females in view of the back support. These may be partly due to differences in the upper body mass of the two genders apart from its distribution. The mass of thorax in males is particularly higher than that in females, while the abdominal masses of males and females are comparable [39]. The mean thorax mass of male and female participants in the present study were estimated as 16.0 kg and 10.2 kg, respectively. The present study shows a considerably greater reduction in peak fore-aft magnitude and corresponding frequency for the male subjects as compared with the female subjects when the sitting condition is changed from NB to WB condition (Table 3). The higher thorax mass of the male subjects may partly account for relatively greater reduction in the peak fore-aft magnitude and primary resonance frequency.

The results in the present study show that increasing the excitation magnitude yield significant ($p < 0.001$) reduction in the primary resonance frequency, which is suggestive of softening tendency of the human body coupled with an elastic seat (Figure 6 and Tables 4 and 5). Such softening trend has been widely reported in many studies that considered body seated on a rigid seat [24,25,40,41]. For both genders, the softening effect was more evident for both genders when the excitation level was increased from 0.25 to 0.50 m/s$^2$. A further increase in the excitation magnitude from 0.50 to 0.75 m/s$^2$, however, showed considerably smaller softening effect. This tendency has also been reported in earlier studies [24,25] considering rigid seats. For the vertical STHT responses, the softening effect was relatively greater for the male subjects than for the female subjects. The greater softening effect on the male subjects may be due to higher skeletal muscle mass and lower body fat mass (lean body mass: 61.6 kg; body fat mass: 19.8 kg) as compared with the female subjects (lean body mass: 41.6 kg; body fat mass: 18.6 kg), which is visible in Table 1. Muscles, owing to their visco-elastic behavior, exhibit thixotropic behavior, where the effective viscosity decreases under a higher rate of shear deformation. The body fat (adipose tissue), on the other hand, exhibits anti-thixotropic behavior, where the viscosity increases with increase in shear rate [42]. The softening effect for the fore-aft STHT responses was relatively greater with the WB condition than that obtained with the NB condition for both genders. Similarly, peak fore-aft STHT magnitude reduction was relatively greater with the WB condition than with the NB condition for both genders. This trend shows the coupling effect of sitting conditions on the fore-aft STHT peak magnitudes and resonance frequencies.

*4.2. Effect of Gender Coupled with Mass-Related Anthropometry*

The body mass effect on the vertical and fore-aft STHT responses is evident, particularly for the primary resonance frequencies. Moreover, the body mass effect is coupled with the sitting condition (Figures 7 and 8; Table 5). Higher stiffness-to-mass ratio of light weight subjects is likely the reason for relatively higher primary resonance frequency of light subjects. The effect of body mass is more pronounced for males compared to females, which is likely due to greater variation in mean body masses of male subjects' groups (60.1, 81.6, and 96.7 kg) than the female subjects' groups (50.4, 61.0, and 69.1 kg) presented in Table 2. The body mass effects on the STHT responses have also been reported in the previous study [14]. The trend in the present study is in line with the reported studies. The results also showed similar effects of mass-related parameters, namely, BMI, body fat mass, and lean body mass on the primary resonance frequencies observed from the vertical and fore-aft STHT (Figures 9 and 10). The effects, however, were not clearly evident for the female subjects. This may be due to the narrow range of the body dimensions. The selected mass-related parameters are highly correlated with the body mass ($r^2 > 0.7$), which may partly be the reason for the same effect of the BMI, body fat mass, and lean body mass as the body mass.

The studies reporting the gender effect on the biodynamic responses to WBV have not considered the contributions and possible coupling with body mass and anthropometric dimensions. Studies reporting gender effects on the STHT responses have reported somewhat inconsistent findings [20]. Some studies have concluded that the gender effect was mostly insignificant [29,43,44], while others show notable gender effect [28]. Such inconsistent findings may be partly due to broad differences in body mass and anthropometric dimensions of the male and female subject populations. The coupled effects of body mass and anthropometry on the gender effect may be minimized by considering male and female subjects of comparable body mass and anthropometric dimensions. This study attempts to evaluate the gender effect by selecting male and female subjects with comparable body mass and anthropometric dimensions.

Comparisons of vertical and fore-aft STHT responses of males and females within two comparable body masses (G60 and G70) showed higher primary resonance frequencies of male subjects compared with the female subjects for both the back support conditions

(Figure 11). Differences in the primary resonance frequencies between two genders of comparable body mass may be due to variation in the skeletal muscle mass and body fat mass between the two genders. The mechanical properties of fat, muscle, and bones differ [45]. Males possess higher muscle masses but lower body fat masses compared to female subjects. In the present study, the lean body mass of males within the G60 and G70 groups were 49.6 and 58.8kg, respectively, while those of females in the respective groups were 41.9 and 47.6 kg. A higher muscle mass has a higher amount of titin, which leads to higher stiffness. The stiffness-to-mass ratio of males is thus relatively higher compared to females, which is likely the reason for higher resonance frequency of males. Dewangan et al. [14] also reported relatively higher primary resonance frequency of males on the basis of STHT responses measured with a rigid seat.

The vertical and fore-aft STHT responses of the male and female subjects with comparable mass-related parameters (BMI, body fat mass, and lean body mass) clearly show strong effects of body fat mass and lean body mass (Figure 12). Dewangan et al. [14] reported that the vertical and fore-aft STHT responses of the male and female subjects with comparable body fat mass are very similar up to nearly 10 Hz, although the finding was based on STHT responses obtained with a rigid seat. The results in this study show considerably higher vertical STHT magnitudes in the vicinity of the primary resonance frequency than the male subjects. A higher peak magnitude for the females could be coupled with the effect of body mass and elastic seat. The elastic seats yield a significantly higher peak vertical STHT magnitude as compared with the rigid seat [11].

*4.3. Effect of Gender and Build-Related Anthropometry*

Higher body-seat contact area resulting from higher hip circumference of subject resulted in lower primary resonance frequency of the vertical and fore-aft STHT responses for both the genders (Figures 13 and 14). A higher hip circumference and body–seat contact area may provide stability to the human body and thus an increase in these two body dimensions may be the reason for lower primary resonance frequency. Wu et al. [46] also observed that the increase in contact area when the seat was changed from rigid to elastic caused a reduction in the primary resonance frequency. Higher correlations between the body mass and hip circumference ($r^2 > 0.91$ for the male subjects and $r^2 > 0.73$ for the female subjects) and body mass and seat contact area ($r^2 > 0.55$ for the male subjects and $r^2 > 0.43$ for the female subjects) may also be attributed to this effect. Although the three ranges of mean hip circumference and the contact area are generally comparable for the male and female subjects, these are substantially higher for females when their proportion to body mass is considered. The effect of hip circumference on the primary resonance frequency was relatively small for the females compared with the males (Figures 13 and 14), which is likely due to relatively larger hip circumference and contact area for the female subject in proportion to the body mass. Furthermore, the pelvis of a female is wider and rounded [31] and may be one of the causes inducing lower lumbopelvic stability in females [32]. The results of the present study on the effect of the hip circumference and contact area are in line with those of Dewangan et al.'s [14]. The primary resonance frequency is comparable for the three ranges of mean contact pressure for the female subjects, while the male subjects showed higher resonance frequency for lower mean contact pressure. This trend of the result is not in line with the data in Dewangan et al.'s work [14]. Variation between the two studies may be due to the type of seat and limitations of the pressure measurement system. Dewangan et al. [9] have reported the limitations of the pressure measurement system. The buttock tissue of a subject is deformed when sitting on a rigid seat, while buttock tissue and elastic seat are deformed when sitting on an elastic seat. Body build has an influence on pressure distribution at ischial tuberosities and thighs on the seat [5]. Thinner subjects show higher pressure in the vicinity of ischial tuberosities, while heavier subjects show relatively higher pressure under the thigh. A similar trend of pressure distribution on automotive seats has been reported by Gyi and Porter [47]. Owing to

observed differences in mean contact pressure distribution, its effect on STHT response could not be clearly identified.

Responses obtained with female subjects show relatively higher peak vertical STHT magnitudes but lower primary resonance frequency when compared to those for males with comparable build-related anthropometric parameters, namely, hip circumference, body–seat contact area, and mean contact pressure (Figure 14). The peak fore-aft STHT magnitude, however, was lower for the female subjects, while the corresponding frequency was comparable with that of the male subjects. On the basis of STHT responses obtained with a rigid seat, Dewangan et al. [14], reported slightly higher peak vertical and fore-aft STHT magnitudes and corresponding frequencies for the males compared to those of females with comparable hip circumference and mean contact area. The size and shape of the pelvis in addition to lower lumbopelvic stability were believed to be likely contributors to lower primary resonance frequency for the females. The primary resonance frequency for the female subjects is also attributable to differences in the body fat mass of the two gender groups, as well as to the coupled effects of the body fat mass and build. For comparable hip circumference, the female subjects possess considerably lower body mass (60.8 kg) and relatively higher body fat mass (19.0 kg) as compared to the male subjects (body mass: 81.3 kg; body fat: 16.7 kg). The hip circumference is positively correlated with body mass and body fat mass for both genders with $r^2$ values ranging from 0.73 to 0.92. Slightly higher body fat mass in females as compared with males may be the reason for the slightly lower primary resonance frequency for the female subjects. Furthermore, the body fat (adipose tissue) is deposited mostly in the pelvis and thighs in females and thus a more uniform contact pressure on the seat occurs, as compared to the male subjects. The results of this study also showed more uniform body-seat pressure distribution for females compared to males Moreover, relatively lower body mass of females together with larger hip circumference causes lower mean contact pressure. The relatively lower resonance frequency observed for females is likely attributable to their lower and more uniform mean contact pressure. There is no gender effect for the vertical STHT magnitude above 10 Hz when a comparable hip circumference and the contact area are considered.

### 4.4. Effect of Gender and Stature-Related Anthropometry

No clear trend in the vertical and fore-aft STHT responses is evident when the three ranges of the stature-related parameters (stature, sitting height, and C7 height) of the two genders are compared (Figures 15 and 16). This trend is similar to that reported by Dewangan et al. [14]. Generally, the primary resonance frequency of the vertical STHT responses also changed significantly between two gender groups with comparable sitting height. The peak vertical STHT magnitudes of the female subjects are greater than those of the male subjects for stature-related parameters. The differences in the peak magnitude and corresponding frequency between the male and female subjects may be due to anatomical differences in the upper body of the male and female subjects. A combination of bending and rocking of the spine is involved in the principal resonance of the human body [48]. Sandover and Dupuis [49] suggested that bending in the lumbar spine and possibly the rocking motion of the pelvis are involved in the principal resonance. The spine of a female tends to be more flexible and is more lordotic than that of a male [50]. The trunk muscle geometry of females and males is also different. A male exhibits larger anatomical cross-sectional areas for trunk muscles (latissimus dorsi, erector spinae, rectus abdominis, external oblique, internal oblique, psoas major, and quadratus lumborum) at most levels (T8 through S1), and at all levels for the vertebral body and torso cross-sectional areas [31]. The flexible spine and lower trunk muscle mass of the female subjects may be the reason for the lower primary resonance frequency.

## 5. Conclusions

The seat-to-head vibration transmissibility of 58 (31 male and 27 female) subjects seated on an elastic seat was measured under two sitting conditions and three magnitudes of random vibration. The results reveal a significant gender effect on the vertical and fore-aft STHT responses, which were strongly coupled with various anthropometric dimensions. The vertical and fore-aft STHT responses of the male and female subjects of comparable body mass were observed to be distinctly different. The primary resonance frequency observed from vertical STHT responses was notably higher for male subjects than the female subjects. The peak STHT magnitudes, however, were similar for the two genders, where the datasets were limited to males and females of comparable body mass. The gender effect was also observed to be coupled with the back support and excitation conditions. Sitting against a vertical backrest resulted in lower vertical primary resonance frequency when compared to that obtained without a back support. This reduction was greater for the females than for the male subjects, while the opposite trend is obtained for the peak fore-aft magnitude and corresponding frequency. The male subjects show more softening effects as compared with the female subjects for the vertical STHT responses, irrespective of the sitting conditions. The primary resonance frequency of the male and female subjects with comparable body fat mass is nearly the same. The vertical STHT magnitude of two genders with the same lean body mass is comparable. The peak fore-aft STHT magnitudes of the male subjects are greater than those of the female subjects of comparable anthropometric parameters except for body mass.

**Author Contributions:** Y.Y., methodology, software and formal analysis, data curation, funding acquisition, writing—original draft preparation; K.N.D., validation, investigation, resources, writing—review and editing; S.R., supervision, project administration, funding acquisition. All authors have read and agreed to the published version of the manuscript.

**Funding:** This research was funded by the National Natural Science Foundation of China grant number 52105114 and the National Foreign Expert Program grant number G2022013006.

**Data Availability Statement:** Not applicable.

**Acknowledgments:** The authors are grateful for the funding support from the National Natural Science Foundation of China (Project No.: 52105114) and the National Foreign Expert Program (Project No.: G2022013006).

**Conflicts of Interest:** The authors declare no conflict of interest.

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
