# Peer review of "Gender and Anthropometric Effects on Seat-to-Head Transmissibility Responses to Vertical Whole-Body Vibration of Humans Seated on an Elastic Seat"

_vibration, doi:10.3390/vibration6010012_

Round 1

Reviewer 1 Report

This study presents an experimental investigation of the effects of gender and ten different anthropometric parameters on the transmission of vertical seat vibration to the head of the body seated on an elastic seat. Biodynamic response characteristics measured in terms of vertical and fore-aft STHT responses of the human body exposed to vertical WBV show a strong gender effect, which is further coupled with the sitting and excitation conditions. Among the various anthropometric dimensions considered, the body mass and fat mass revealed strong influences on the primary resonance frequency. However, the manuscript has to be revised according to the following comments.

1.       There are too many abbreviations in the manuscript, especially in the abstract. These abbreviations easily confuse the readers.

2.       Only experimental investigation was conducted on the biodynamic response. Due to insufficient subjects, the obtained results are not convinced. Theoretical analysis of biodynamic response is recommended to be conducted.

Author Response

Please kindly check the attachment. 

Reviewer 2 Report

This paper examines if and how differences in morphometry of the body between males and females affect the resonant frequency of the torso and fore-aft motion of the head in response to whole body vibration.  It also examines how a hard vs cushioned seat affects these measures.  This study is because more women are entering occupations where they are exposed to WBV and understanding how their body responds, and how different seats affect this response, will help manufacturer's design seats to protect both men and women.

The study is well designed and for the most part the paper is well written.  However, there are some places that need editing along with some of the figures:

1.  When talking about males vs female, especially with reference to morphometrics and kinematics, it’s more appropriate to use the term sex.  Gender refers to what a person sees them self as regardless of their physical or genetic sex.  I would change gender to sex unless the journal prefers the use of the word gender.

Introduction

Page 1, lines 39 and 40. Instead of invariably the authors may want to consider “predominantly or mostly, and then the “with a few exceptions” can be deleted

Page 2, line 55, the authors may want to consider saying “impact vibration” or” impact in the vertical direction” instead of impacts type of motions.

Page 2, line 65  Minor edits to this sentence would make it easier to understand “These differences  among the reported data, in addition to large inter-subject variability observed in individual studies, are partly attributable to variations in physical characteristics of the subjects”

Page 2 line 68:  The authors consider editing for clarity “Male and female populations, in particular, exhibit a range of differences in their physical characteristics”.

Page 2 line 71.  Do the authors mean the skeleton of females is round or the pelvic region?

Page 2 line 73, instead of saying higher the authors should consider saying “greater”

Page 2 line 87-88:  This sentence does not make sense “Goggins et al. [19] assessed seat-to-head transmissibil-87 ity and self-reported discomfort of 5 females and 6 males with different postures and 88 seated conditions, while the gender effect on the STHT responses was not investigated.”

Page 3 line 123: The authors use the past tense in this paragraph.  Please change the word “are” to “were”

Page 3, line 131-132 please clarify that the protocols or procedures were approved.

Page 3: line 133 please define “C-7”  was

Page 4 table 1:  Do the first two numbers in the table represent the range?  If so, please add this to the table caption

Page 4, line 151, please define SAE

Page 4, line 152: The authors may want to consider editing “Although the static force-deflection characteristics of the elastic 152 seat revealed nonlinear dependence on the applied load, the static stiffness corresponding 153 to 330, 440, and 530 N applied loads were obtained as 20.6, 34.0, and 40.7 kN/m” to say “…530 N measured applied loads  were 20.6….)”

Page 4 line 155, do the authors mean standing or seated?

Page 4, line 163 M. Pranesh?

Page 4 line 164 : consider editing to say, “The WBVVS was comprised of…”

Page 5 line 179: Do the authors mean the elastic seat not only alters transmissibility of the platform vibration, but is nonlinearly dependent…?

Page 6, line 209, would it be more accurate to say that “Additional feet support was used to ensure that the subjects knees were bent at 90º with their thighs were parallel to the platform and their lower legs perpendicular to the platform”?

Table 2.  How were the levels for grouping determined? The authors should to define G60 and G70 in the text before the table or in the caption for the table.

Please add letters to the graphs in your figures.

Figure 5.  The text says that the fore-aft magnitude is higher in the condition with the back support in males.  However, in the females it looks like the vertical magnitude is greater with back support in females, and this is not mentioned in the text or in the figure caption.

Page 16, line 466, beginning of the sentence “was performed” instead of “is performed”

Page 16, line 489 “the female subjects had considerably higher …”

Page 26 Line 660, If muscle mass is lower in females, overall there will be less of the protein titan which plays a role in passive muscle stiffness.  However, this does not mean muscle stiffness was lower in females because it’s not the overall level of the protein “titan”, but the ratio of titan to muscle mass that is important.  Please edit to reflect this

Page 27 lines 712-713:  The following sentence does not make sense “The higher thorax mass of the 712 male subjects may partly be attributed to a greater reduction in the peak fore-aft magnitude and primary resonance frequency.”  Do the authors mean “The higher thorax mass of the may subjects may in part account for the greater reduction in …”

Page 28, line 725, please edit to say “which is evident”

Page 28 Line 727, do the authors mean making the response of the muscle shear rate dependent?

Page 28 Line 731 please change reduced to reduction was

page 28,  line 737  There is something missing or that doesn’t make sense with the following sentence” The body mass effect on the vertical and fore-aft STHT responses is evident, particularly for the primary resonance frequencies and the body mass effect is coupled with the 737 sitting condition”

page 28 line 754 do they suggest inconsistent findings or do they report inconsistent findings between studies?

Page 28 lines 758 – 761 Do you think this was true?  “The coupled effects 758 of body mass and anthropometry on the gender effect may be minimized by considering 759 male and female subjects of comparable body mass and anthropometric dimensions. This 760 study attempts to evaluate the gender effect by selecting male and female subjects with 761 comparable body mass and anthropometric dimensions.”  There were still differences in the responses, even when the investigators tried to control for a number of subject-dependent factors. 

Page 29, line 772.  As mentioned above (line 660) the following statement is not necessarily true “The coupled effects of body mass and anthropometry on the gender effect may be minimized by considering male and female subjects of comparable body mass and anthropometric dimensions. You would need to measure titan in an equivalent amount of muscle, and if titan is lower if lower in females than males, you may be able to say this.  In contrast, you could just say that because there is a lower muscle mass in females, stiffness contributed to muscle mass is reduced.

Page 30, line 880: buttock does not need to be capitalized

Page 30 line 816:  The authors may want to consider saying “This difference in pressure may be the reason the …”

Page 30 lines 819 – 824:  The following sentence ” Comparisons of the STHT responses of two gender groups with comparable build- related anthropometric parameters (hip circumference, body-seat contact area, and mean contact pressure) show higher peak vertical magnitude and lower primary resonance frequency for the female subjects as compared with the male subjects (Fig. 14), while relatively lower peak fore-aft magnitude and comparable primary resonance frequency for female subjects than male subjects.”  should be made into two sentences (put a period after Fig 14).  And the second part of the sentence beginning “while relative…” does not make sense. 

Page 30 line 837 please change thighs to thigh or delete the word regions.

Page 31. Lines 858 and 859 to say that the spine of a female is more arched and more lordotic is somewhat redundant you could probably delete the sentence beginning “a female also tends…” and say “The spine of a female tends to be more flexible and the lumbar region is more lordotic”

Page 31 line 860 please the word “the” before the word “larger”

Page 31 line 871: please change “is” to “were” and instead of coupled the authors might want to say associated with

References:  the authors should check their reference.  In some of them, the manuscript title is in all capital letters while in others it is not.

Author Response

Please kindly check the attachment. 

Reviewer 3 Report

The paper presents a very rich set of experimental data related to measurements of seat-to-head transmissibility in presence of whole-body vibrations.

The experimental set-up is clearly described and many cases are considered, in order to show the effect of multiple parameters (gender, body mass, presence or absence of back support, etc.) on the dynamic behavior of the system while underlining the separate effect of each of them, too. Data are clearly analysed and interesting observations are given in the final discussion.

The possible unconsistency that could be produced by measurements with a narrow scatter (for example if good quality and repeatibility of measures were not guaranteed) seems here to be avoided, even through the repetition of measurements and the following use of in-depth statistical analyses. That is, the application of a good scientific approach. 

Comments always appear very "sincere", since even limitations are explained when needed.

A rich scientific bibliography is given and previous researches are analyzed in order to clearly show the  innovative aspects of the study. Every obtained result is compared with other ones previously described in international literature, that is appreciable.

English is good and the organization of the manuscript is clear.

For these reason, I think that the paper could be published after applying only some minor revisions, that is:

- the abstract is very long and detailed, it is very similar to (or even better than) the Conclusion paragraph. In my opinion, abstract should be reduced and details about results should be mainly concentrated within the Conclusions.

- a schematic representation of "human body + seat + vibration directions + sensors" should be added at the beginning of the paper, for example adding a second picture to Figure 1.

- pag.5, line 180: the sentence is not completely clear to me. Maybe a subject as "its transmissibility" or "its dynamic behavior" should be added before "is also nonlinearly dependent".

- pag. 25, line 606: please check English? "that" ?

- pag. 29, lines 791-793: could you please justify better the idea of a connection between dimensions of the seat-body contact (increased stability) and the reduced primary frequency? Similarly, please explain better the concept expressed at lines 840-842 pag.30.

- The bibliography is very rich, even if maybe the "old" papers [1970-2000] are dominant with respect to more recent ones. In any case, it seems to me that even the recent literature is complete. I only sugget to add and comment the following paper (complementary to another one that it is already cited):

Bhiwapurkar et al., "Seat to Head Transmissibility during Exposure to Vertical Seat Vibration: Effects of Posture and Vibration Magnitude", International Journal of Acoustics and Vibration, Vol. 24, No. 1, 2019 (pp. 311), https://doi.org/10.20855/ijav.2019.24.11108

Author Response

Please kindly check the attachment. 

Round 2

Reviewer 1 Report

The revised manuscript can be accepted by the Journal of Vibrtion.